# Stop the Flip-Flop: Context-Preserving Verification for Fast Revocable Diffusion Decoding

Yanzheng Xiang [* 1]   Lan Wei [* 2]   Yizhen Yao [1]   Qinglin Zhu [1]   Hanqi Yan [1]   Chen Jin [3]   Philip Alexander Teare [3]
Dandan Zhang [2]   Lin Gui [1]   Amrutha Saseendran [3]   Yulan He [1 4]

## Abstract

Parallel diffusion decoding can accelerate diffusion language model inference by unmasking multiple tokens per step, but aggressive parallelism often harms quality. Revocable decoding mitigates this by rechecking earlier tokens, yet we observe that existing verification schemes frequently trigger flip-flop oscillations, where tokens are remasked and later restored unchanged. This behaviour slows inference in two ways: remasking verified positions weakens the conditioning context for parallel drafting, and repeated remask cycles consume the revision budget with little net progress. We propose **COVER** (Cache Override Verification for Efficient Revision), which performs leave-one-out verification and stable drafting within a single forward pass. COVER constructs two attention views via KV cache override: selected seeds are masked for verification, while their cached key value states are injected for all other queries to preserve contextual information, with a closed form diagonal correction preventing self leakage at the seed positions. COVER further uses stability-aware adaptive verification to prioritise high-risk seeds and adjust the verification budget at each step, thereby reducing unnecessary revisions and enabling faster decoding while preserving output quality across benchmarks.

## 1. Introduction

Autoregressive language models (Abhimanyu Dubey et al., 2024; Brown et al., 2020; Radford et al., 2019; Radford &

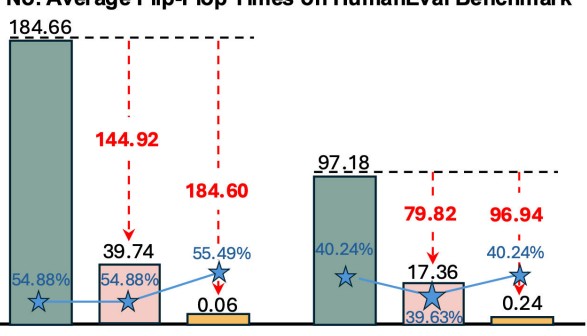

*Figure 1.* Flip-flop behaviour on HumanEval for Dream-Ins-7B and LLaDA-Ins-8B under two revocable baselines (Saber, WINO) and ours (COVER). Unlike baselines that repeatedly ReMask, COVER uses context-preserving in-place verification to reduce oscillatory revisions while maintaining generation quality.

Narasimhan, 2018) generate text token by token and remain the dominant paradigm for high quality generation. Yet this sequential decoding is a persistent inference bottleneck, and early errors can propagate through the remainder of the output (Valmeekam et al., 2023; Stechly et al., 2023). Diffusion large language models (dLLMs) offer an appealing alternative: they denoise an initially masked sequence and can, in principle, update many positions in parallel (Li et al., 2022b). In practice, however, aggressive parallel unmasking often harms generation quality (Hong et al., 2025; Dong et al., 2025), so dLLMs frequently revert to conservative decoding that unmasks only one position per step, largely sacrificing the promised speed gains (Nie et al., 2025; Ye et al., 2025; Xie et al., 2025).

Recent work attempts to bridge this gap with revocable parallel diffusion decoding. These methods draft multiple tokens in parallel and then revisit a subset of previously unmasked positions using the newly available context, optionally revoking them by resetting to `[MASK]`. WINO (Hong et al., 2025) performs verification through an auxiliary shadow block, whereas Saber (Dong et al., 2025) triggers remasking based on confidence drops. Although revocation im-

---

[*]Equal contribution  [1]King's College London, UK [2]Imperial College London, UK [3]Centre for AI, Data Science & Artificial Intelligence, BioPharmaceuticals R&D, AstraZeneca, UK [4]The Alan Turing Institute, UK. Correspondence to: Yulan He <yulan.he@kcl.ac.uk>.

*Proceedings of the 43rd International Conference on Machine Learning*, Seoul, South Korea. PMLR 306, 2026. Copyright 2026 by the author(s).

proves robustness, existing verification mechanisms introduce substantial overhead. WINO increases effective sequence length and memory footprint, and both methods depend on explicit remasking, which replaces content tokens with [MASK] for all queries and can destabilise subsequent drafts, leading to slower net denoising progress.

In this work, we highlight an inefficiency of revocable decoding that standard accuracy metrics do not capture. We observe flip-flop oscillations, where a position is remasked and later re-unmasked to exactly the same token. Figure 1 shows that such oscillations occur frequently under existing revocable baselines across dLLMs, indicating that many verification actions consume iterations without producing a correction. This creates two coupled inefficiencies. First, remasking replaces a content bearing embedding with [MASK], weakening the conditioning context used by other positions during parallel drafting. Second, each ineffective remask spends future unmask budget merely to restore the same token, reducing net denoising progress under any fixed step or unmask budget.

To solve this, we propose **COVER**[1] (Cache Override Verification for Efficient Revision), a context-preserving single-pass verification mechanism for revocable parallel diffusion decoding. At each step, COVER rechecks a small seed set by masking these positions in the input while overriding their KV states with cached values from the previous step. This dual view computation keeps the drafting context for all non seed queries unchanged, yet enables faithful leave one out verification on the seeds via a diagonal correction that removes self leakage. To make cache reuse reliable, COVER chooses seeds using a stability aware score that trades off uncertainty against estimated influence on the remaining masked positions, adapts the verification token number per step, and updates verified positions by KEEP, REPLACE, or REMASK to avoid ineffective remasking cycles.

Our contributions are as follows:

- We identify flip-flop oscillations as a dominant inefficiency in revocable diffusion decoding and show how explicit remasking weakens drafting context and wastes the revision budget.

- We introduce an in place KV cache override verification mechanism with diagonal correction, enabling faithful leave-one-out checks and stable parallel drafting within a single forward pass.

- We propose stability aware and adaptive seed selection that prioritises uncertain and influential positions while avoiding unstable cache reuse, enabling efficient multi-token verification.

[1]Our code is available at: https://github.com/xyzCS/COVER

- We show that COVER generally preserves or improves accuracy while substantially reducing decoding steps, yielding consistent end-to-end speedups of up to $11.64\times$ (Dream-Ins-7B), which supports reliable multi-token drafting via context-preserving in-place verification.

## 2. Related Work

**Diffusion Large Language Models (dLLMs).** Diffusion language models generate text by iteratively denoising a partially masked sequence, enabling multi token generation in principle. Early work studied both continuous diffusion for text (Li et al., 2022a; Gong et al., 2022; Han et al., 2023) and discrete formulations (Ou et al., 2024; Lou et al., 2023; Austin et al., 2021a; Sahoo et al., 2024). Among these, masked discrete diffusion models have proven most amenable to large scale training and deployment (Sahoo et al., 2024). Recent releases include open models such as LLaDA (Nie et al., 2025) and Dream (Ye et al., 2025), as well as commercial systems such as Mercury (Labs et al., 2025) and Gemini Diffusion (Deepmind, 2025). Despite their potential, practical inference remains challenging: aggressive parallel unmasking often degrades generation quality, while bidirectional attention and the lack of a stable KV cache make each decoding step expensive. Closing this gap between multi token capacity and reliable fast inference is an active research direction.

**dLLMs Acceleration.** Existing acceleration methods mainly follow two directions: reducing per step compute via KV reuse (Liu et al., 2025; Wu et al., 2025; Song et al., 2025) and reducing the number of steps via parallel decoding (Israel et al., 2025; Wang et al., 2025b; Kang et al., 2025). On the systems side, Fast dLLMs (Wu et al., 2025) observes that KV states change smoothly across diffusion steps under full attention and proposes to cache and update them blockwise, amortising recomputation. On the algorithmic side, parallel decoding unmasks multiple positions per step, typically guided by confidence criteria, and relies on verification with optional remasking to correct erroneous drafts (Wang et al., 2025a; Kong et al., 2025; Dong et al., 2025; Ye et al., 2026). WINO (Hong et al., 2025) performs verification using an auxiliary shadow block with a stricter criterion than drafting, which improves selectivity but introduces additional computation. dParallel (Chen et al., 2025) combines self-distillation with entropy threshold-based remasking to reduce steps, but it requires retraining the diffusion model to obtain the high certainty drafts needed for aggressive parallel unmasking. Instead, COVER achieves faithful leave-one-out verification in place via KV cache override with diagonal correction, and selects verification seeds with a stability aware rule, enabling fast parallel decoding without extra blocks or retraining.

## 3. Revocable Parallel Diffusion Decoding

Let $\mathcal{V}$ be a vocabulary and let [MASK] be a special token. We consider conditional generation with a prompt $X$ and a response of fixed length $L$. At step $t \in \{0, \ldots, T\}$, the partial state is $Y^{(t)} = (y_1^{(t)}, \ldots, y_L^{(t)}) \in (\mathcal{V} \cup \{[\text{MASK}]\})^L$ with $Y^{(0)} = [\text{MASK}]^L$. We denote masked and unmasked indices by $\mathcal{M}_t := \{i \in [L] : y_i^{(t)} = [\text{MASK}]\}$ and $\mathcal{U}_t := [L] \setminus \mathcal{M}_t$. Given $(X, Y^{(t-1)})$, the diffusion model outputs per-position token distributions $\{p_\theta^{(i)}(\cdot \mid X, Y^{(t-1)})\}_{i=1}^L$.

**Decoding protocol.** Revocable parallel diffusion decoding iterates for $t = 1, \ldots, T$. Step $t$ takes as input the current state $Y^{(t-1)}$ and a seed set $\mathcal{S}_{t-1} \subseteq \mathcal{U}_{t-1}$ (with $\mathcal{S}_0 = \emptyset$), where $\mathcal{S}_{t-1}$ contains previously unmasked positions scheduled to be rechecked at step $t$. The procedure is specified by three step-dependent rules: a drafting rule (choose $\mathcal{D}_t$), a verification rule (produce updates and a remask set), and a seed selection rule (choose $\mathcal{S}_t$).

**Drafting.** A drafting rule selects a set $\mathcal{D}_t \subseteq \mathcal{M}_{t-1}$ of currently masked positions to unmask in parallel. For each $i \in \mathcal{D}_t$, it proposes a token $\hat{y}_i^{(t)} := \arg\max_{v \in \mathcal{V}} p_\theta^{(i)}(v \mid X, Y^{(t-1)})$. A typical instantiation ranks masked positions by confidence $c_i^{(t-1)} := \max_{v \in \mathcal{V}} p_\theta^{(i)}(v \mid X, Y^{(t-1)})$ and selects the top ones, optionally subject to a budget $|\mathcal{D}_t| \le B$.

**Verification with optional revocation.** Given the newly drafted context, a verification rule revisits each seed position $i \in \mathcal{S}_{t-1}$. It either outputs an updated token $\bar{y}_i^{(t)} \in \mathcal{V}$, or revokes the position by resetting it to [MASK]. The revoked indices form the remask set $\mathcal{R}_t \subseteq \mathcal{S}_{t-1}$.

**State update.** Initialize $Y^{(t)} \leftarrow Y^{(t-1)}$, then apply the drafting and verification outcomes:

$$
y_i^{(t)} = \begin{cases} \hat{y}_i^{(t)}, & i \in \mathcal{D}_t, \\ [\text{MASK}], & i \in \mathcal{R}_t, \\ y_i^{(t-1)}, & \text{otherwise.} \end{cases}
$$

Equivalently, $\mathcal{U}_t = (\mathcal{U}_{t-1} \cup \mathcal{D}_t) \setminus \mathcal{R}_t$ and $\mathcal{M}_t = [L] \setminus \mathcal{U}_t$.

**Seed selection.** A seed selection algorithm chooses the next seed set $\mathcal{S}_t \subseteq \mathcal{U}_t$, which will be verified at step $t + 1$. Decoding terminates when $\mathcal{M}_t = \emptyset$ or when a step budget is reached.

## 4. Flip-Flop Oscillations

Revocable decoding improves quality by allowing previously unmasked tokens to be remasked and refined under richer context. However, in practice we observe a pathological behavior that we call flip-flop oscillations, which can substantially slow down inference without providing meaningful corrections.

**Definition.** During revocable diffusion decoding, a position can be unmasked, later remasked to [MASK], and then unmasked again. Fix a position $i$ and record the token predicted each time $i$ transitions from [MASK] to a concrete token. We say a flip-flop event occurs at position $i$ if two consecutive such unmaskings predict the same token, meaning that the intermediate remask does not change the model's discrete decision. Equivalently, a flip flop corresponds to an ineffective remask action that is later undone by restoring the same token. Let $F_i$ denote the number of flip flop events at position $i$, and define the total flip flop count for the sequence as $F = \sum_{i=1}^L F_i$. In our empirical study, flip flops dominate revocation in existing methods: around 99% of Saber's ReMask operations are ineffective, and for WINO the ineffective fraction remains close to 90% across datasets (Section 6.3).

**Flip-flop oscillations slow down decoding.** Flip-flop oscillations reduce efficiency even when revocation rarely changes the final discrete prediction. We highlight two sources of overhead:

1. **Remasking weakens the conditioning context for parallel drafting.** When a previously unmasked token is reset to [MASK], the input replaces a content bearing embedding with an uninformative placeholder, so other positions that attend to it temporarily lose semantic signal. Empirically, many revoked positions are later repredicted as exactly the same token, which means this transient context deletion often provides no corrective benefit. Nevertheless, it still lowers confidence at remaining masked positions, typically shrinking the drafted set $\mathcal{D}_t$ and slowing the net rate at which new tokens can be committed.

2. **Remasking consumes the decoding budget and reduces net progress.** From the state update $\mathcal{U}_t = (\mathcal{U}_{t-1} \cup \mathcal{D}_t) \setminus \mathcal{R}_t$, the net expansion per step is $|\mathcal{U}_t| - |\mathcal{U}_{t-1}| = |\mathcal{D}_t| - |\mathcal{R}_t|$. Each flip-flop event increases $|\mathcal{R}_t|$ without producing a new assignment and forces a later step to spend an unmask slot merely to restore the same token, thereby wasting iterations under any fixed step or unmask budget. We formalize this overhead in Appendix A (Lemma A.1), proving that any additional remask or flip-flop event increases the required number of decoding steps under a fixed per-step unmask budget.

## 5. Method

We propose COVER, an in-place single-pass verification mechanism for revocable parallel diffusion decoding (Figure 2). At each step, COVER performs parallel drafting on masked positions while simultaneously verifying a small seed set from the previous step. Verification is implemented

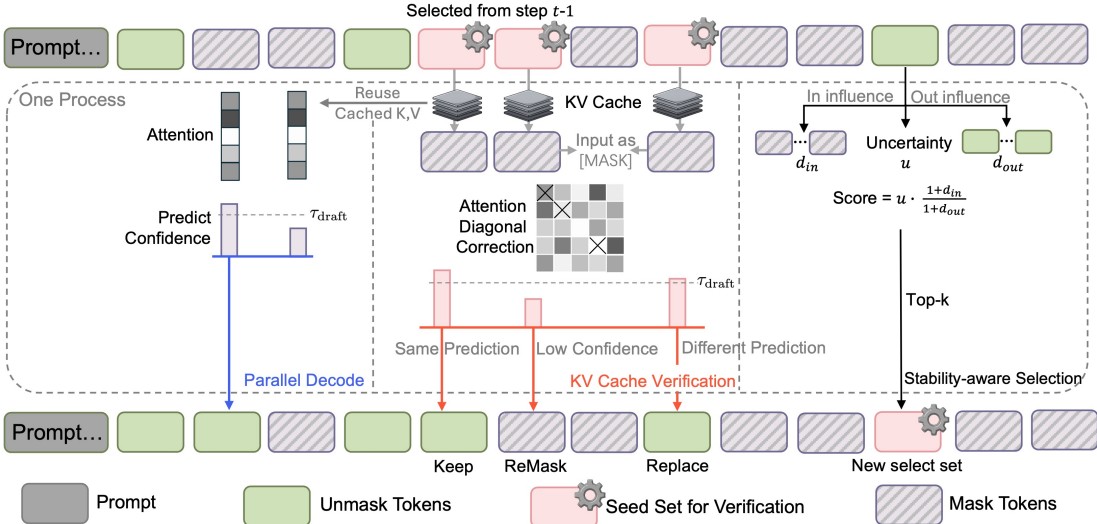

*Figure 2.* Overview of our single-pass revocable diffusion decoding. At step $t$, the model drafts multiple masked positions in parallel and verifies a seed set selected from step $t-1$. Verification masks the seeds in the input but injects their cached $K, V$ states so non-seed queries see an unchanged context. An attention diagonal correction is applied at the masked seed positions to prevent self-leakage and enable re-prediction from the surrounding context. Each seed is then updated by KEEP, REPLACE, or REMASK, and a stability-aware score based on uncertainty and in/out influence selects the next seed set via top-$k$.

by masking the selected positions in the input but overriding their key value states with cached activations, together with a diagonal correction that prevents self-leakage. This dual view computation preserves a stable conditioning context for drafting, and yields faithful leave-one-out checks for the verified positions. Each verified position is then assigned KEEP, REPLACE, or REMASK to avoid ineffective flip-flop cycles. Finally, a stability aware seed selection rule prioritises high-risk positions, and an adaptive revision rate controls how many positions are verified per step.

### 5.1. Dual-View Feed Forward through KV Cache Override

At denoising step $t$, COVER takes as input the current partial state $Y^{(t-1)}$ and a seed set $\mathcal{S}_{t-1} \subseteq \mathcal{U}_{t-1}$ selected at the end of step $t-1$ (Sec. 5.3). Positions in $\mathcal{S}_{t-1}$ are previously unmasked tokens scheduled to be rechecked at step $t$, and the verification rule will optionally output a remask set $\mathcal{R}_t \subseteq \mathcal{S}_{t-1}$.

Our goal is to obtain two types of predictions within a single forward pass: (i) a faithful leave one out style verification distribution at each seed position $i \in \mathcal{S}_{t-1}$, where the model re-predicts $y_i^{(t-1)}$ from surrounding context with the input at $i$ set to [MASK]; and (ii) stable drafting distributions for all non seed positions, including masked positions to be drafted, whose queries should still condition on the same seed representations as in step $t-1$.

**Masked seed input for verification.** We first construct a

verification input by masking only the seed positions:

$$\tilde{y}_j^{(t-1)} = \begin{cases} \texttt{[MASK]}, & j \in \mathcal{S}_{t-1}, \\ y_j^{(t-1)}, & \text{otherwise.} \end{cases}$$

Let $(Q_\ell, K_\ell, V_\ell)$ denote the query, key, and value states computed from $\tilde{Y}^{(t-1)}$ at transformer layer $\ell$.

**KV cache override yields a stable drafting view.** Naively masking $\mathcal{S}_{t-1}$ would delete their information from the context of every other query, weakening parallel drafting. COVER preserves the seed context by overriding only the memory columns at the seed positions with their cached key value states from step $t-1$. Concretely, when $\mathcal{S}_{t-1}$ is selected, we cache the per layer key and value states $\{(\bar{K}_{\ell,j}^{(t-1)}, \bar{V}_{\ell,j}^{(t-1)})\}_{j \in \mathcal{S}_{t-1}}$. At step $t$, we form an overridden memory $(K_\ell', V_\ell')$ by

$$(K_{\ell,j}', V_{\ell,j}') = \begin{cases} (\bar{K}_{\ell,j}^{(t-1)}, \bar{V}_{\ell,j}^{(t-1)}), & j \in \mathcal{S}_{t-1}, \\ (K_{\ell,j}, V_{\ell,j}), & \text{otherwise.} \end{cases}$$

We then run attention once for all positions using this overridden memory:

$$O_\ell^{\text{ovr}} = \text{Attn}(Q_\ell, K_\ell', V_\ell') = \text{softmax}\left(\frac{Q_\ell K_\ell'^\top}{\sqrt{d}}\right) V_\ell',$$

denote the resulting output vector at position $i$ by $o_{\ell,i}^{\text{ovr}}$. Here, $d$ is the key dimension per attention head. For any query position $i \notin \mathcal{S}_{t-1}$, the seed columns in memory are exactly the cached representations from step $t-1$, so non seed queries continue to condition on a stable seed context even though the seed tokens are masked in the input.

**Diagonal correction for faithful verification.** The KV override attention run above is sufficient for stable drafting, but it is not faithful for verifying a seed position $i \in \mathcal{S}_{t-1}$. Although $y_i^{(t-1)}$ is masked in $\tilde{Y}^{(t-1)}$, naively overriding the seed columns would still place the cached pair $(k_i', v_i')$ on the diagonal column $j = i$, creating a direct self conditioning path that can leak the token being verified.

To obtain a leave one out view for each seed query $i$, we keep the overridden columns for all $j \neq i$ but restore the diagonal column to the key and value computed from the masked input:

$$(k_j^{(i)}, v_j^{(i)}) = \begin{cases} (k_i, v_i), & j = i, \\ (k_j', v_j'), & j \neq i. \end{cases}$$

This modification changes only the diagonal attention score in row $i$. However, since attention probabilities are normalized by a softmax, changing the diagonal score also rescales the entire attention distribution in that row. We therefore apply a post-hoc diagonal correction. Let $\alpha_i$ be the diagonal attention weight under the overridden run and let $\delta_i$ be the diagonal score shift after restoring $(k_i, v_i)$. Then the corrected attention weights are obtained by a single row wise rescaling:

$$w_{i,j} = \frac{w_{i,j}^{\mathrm{ovr}}}{r_i} \ (j \neq i), \qquad w_{i,i} = \frac{w_{i,i}^{\mathrm{ovr}} \exp(\delta_i)}{r_i},$$

$$r_i = 1 + \alpha_i\big(\exp(\delta_i) - 1\big).$$

We then update the attention output at $i$ accordingly, while leaving all non seed queries unchanged. The full derivation and implementation details are provided in Appendix B.

### 5.2. Drafting: Multiple Token Unmasking

We adopt the parallel drafting scheme described in Sec. 3. At decoding step $t$, let $\mathcal{M}^{(t)}$ be the set of masked positions, and let $c_i^{(t)}$ denote the confidence score (as defined in Sec. 3) for each $i \in \mathcal{M}^{(t)}$. We draft new tokens by selecting all masked positions whose confidence exceeds a threshold:

$$\mathcal{D}^{(t)} = \big\{ i \in \mathcal{M}^{(t)} : c_i^{(t)} > \tau_{\mathrm{draft}} \big\}.$$

To avoid overly aggressive updates within a single step, which can introduce many errors, we additionally cap the number of drafted positions by a maximum budget $B$. When $|\mathcal{D}^{(t)}| > B$, we keep only the $B$ positions with the largest confidence values.

**Revision outcomes for previously verified positions.** In the same forward pass, the model re-predicts the seed positions $\mathcal{S}_{t-1}$ from the previous step under the verification view. For each $i \in \mathcal{S}_{t-1}$, let $\tilde{y}_i^{(t)} = \arg\max_v p_i^{(t)}(v)$ and $\tilde{c}_i^{(t)} = p_i^{(t)}(\tilde{y}_i^{(t)})$. We apply a three-way revision rule

(KEEP/REPLACE/REMASK):

$$y_i^{(t)} = \begin{cases} y_i^{(t-1)}, & \tilde{y}_i^{(t)} = y_i^{(t-1)}, \\ \tilde{y}_i^{(t)}, & \tilde{y}_i^{(t)} \neq y_i^{(t-1)}, \quad \tilde{c}_i^{(t)} > \tau_{\mathrm{draft}}, \\ \texttt{[MASK]}, & \text{otherwise.} \end{cases}$$

The revision rule is designed to avoid unnecessary revocations. KEEP skips a ReMask when the verified token matches the current assignment, and REPLACE commits a confident correction in place. Both actions reduce the remask set $\mathcal{R}_t := \{i \in \mathcal{S}_{t-1} : y_i^{(t)} = \texttt{[MASK]}\}$, thereby suppressing flip-flop revisions and preserving net progress under a fixed unmasking budget.

Decoding terminates once $Y^{(t)}$ contains no $\texttt{[MASK]}$ tokens; for Instruct models, we stop early when the end-of-sequence token is generated.

### 5.3. Stability Aware Seed Selection and Adaptive Revision Rate

Verifying too many positions in parallel can be harmful when their cached representations are unstable, as overriding such KV states may perturb the predictions of other tokens. We therefore restrict verification to a small, adaptively chosen seed set $\mathcal{S}_t \subseteq \mathcal{U}_t$.

**Stability aware seed scoring.** For each $j \in \mathcal{U}_t$, we score its verification priority by combining (i) risk of being incorrect, (ii) how much the remaining masked positions rely on it as context, and (iii) how likely its cached KV state will drift after the current draft.

Let $A^{(t)} \in [0, 1]^{L \times L}$ denotes the last layer attention matrix (averaged over heads) from the current forward pass. We define three signals:

$$u^{(t)}(j) = -\log p_j^{(t)}\big(y_j^{(t)}\big),$$

$$d_{\mathrm{in}}^{(t)}(j) = \sum_{q \in \mathcal{M}_t} A_{q \to j}^{(t)},$$

$$d_{\mathrm{out}}^{(t)}(j) = \sum_{i \in \mathcal{D}_t} A_{j \to i}^{(t)}.$$

Here, $u^{(t)}(j)$ is the uncertainty of the currently assigned token at position $j$, so larger values indicate higher verification risk. The term $d_{\mathrm{in}}^{(t)}(j)$ measures the downstream influence of $j$ on the not yet generated tokens, namely the total attention mass from still masked queries to $j$. The term $d_{\mathrm{out}}^{(t)}(j)$ measures the draft sensitivity of $j$, namely how strongly $j$ attends to newly drafted positions; a large value suggests that the representation at $j$ is likely to change after drafting, making KV reuse less stable.

We combine them as

$$\text{Score}^{(t)}(j) = u^{(t)}(j) \frac{1 + d_{\text{in}}^{(t)}(j)}{1 + d_{\text{out}}^{(t)}(j)}.$$

Thus, we prioritise seeds that are uncertain and influential, while penalising those whose cached states are likely to drift under the newly introduced context.

**Adaptive revision rate.** Rather than fixing the seed number for revision per step, we adapt it to the empirical score distribution. Let $n_t = |\mathcal{U}_t|$ and $\{s_j\}_{j \in \mathcal{U}_t}$ be the scores. We define the empirical cumulative distribution function (CDF)

$$F_t(s) = \frac{1}{n_t} \sum_{j \in \mathcal{U}_t} \mathbb{I}\{s_j \le s\}.$$

Define the empirical mean $\bar{s}_t = \frac{1}{n_t} \sum_j s_j$ and tail mass

$$\pi_t = 1 - F_t(\bar{s}_t).$$

We then set the verification number as $|\mathcal{S}_t| = \lceil \sqrt{n_t \pi_t} \rceil$, and select the top-scoring positions accordingly. To avoid stale KV reuse and repeated edits, we exclude positions from seed selection if they were selected in the previous step or have reached the per-position ReMask budget $M_{\text{remask}}$.

## 6. Experiment

### 6.1. Experimental Settings

**Implementation Details.** We conduct experiments on four different dLLMs, namely LLaDA-Base-8B, LLaDA-Ins-8B (Nie et al., 2025), LLaDA-1.5-8B (Zhu et al., 2025a), and Dream-Ins-7B (Ye et al., 2025). For consistency and robustness, we set the decoding temperature to zero and greedily unmask the token with the lowest entropy at each step. We adopt the semi-autoregressive sampling strategy (Nie et al., 2025), which segments the output sequence into a set of blocks that are generated sequentially in a left-to-right order. In our evaluation, we set the generation length to 256 and 512, use a block length of 32 for Dream-Ins-7B, and use a block length of 64 for the other models. We fix the per-step drafting budget to $B = 15$, set the per-position REMASK budget $M_{\text{remask}}$ around 5 in practice, and restrict the drafting confidence threshold to the conservative set $\tau_{\text{draft}} \in \{0.7, 0.8, 0.9\}$. All experiments are conducted on four NVIDIA H200 GPUs.

**Datasets.** We evaluate our approach on four benchmarks covering mathematical reasoning and code generation. For mathematical reasoning, we consider GSM8K (Cobbe et al., 2021), which contains grade-school–level word problems, and the more demanding MATH500 (Lightman et al., 2023), composed of competition-style mathematics questions. For code generation, we benchmark on MBPP (Austin et al., 2021b), which focuses on introductory Python programming

tasks, and HumanEval (Mark Chen et al., 2021), a collection of hand-written problems designed to assess program synthesis ability. All "Instruct" variant models are evaluated in the zero-shot setting, while standard few-shot protocols are adopted on the LLaDA-Base-8B model specific to each benchmark: zero-shot for HumanEval, three-shot for MBPP, four-shot for MATH500, and eight-shot for GSM8K (Zhu et al., 2025b).

**Metrics.** To evaluate the effectiveness and efficiency of our approach, we utilise three primary metrics: Acc., Steps, and Speed. For performance assessment, we report standard accuracy on mathematical reasoning benchmarks and the pass@1 rate for code generation tasks. Under deterministic decoding, these scores are computed from a single run over the full benchmark split. Efficiency is quantified by tracking the average number of decoding steps required per sample across the entire dataset. Finally, we measure relative speedup by calculating the ratio of the total inference time: specifically, the total runtime of standard greedy decoding divided by the total runtime of the parallel diffusion decoding methods.

To characterise flip–flop behaviour, we additionally report revision-level statistics: (1) No. Total Revision, the total number of revision operations; (2) No. Eff. Revision, the number of effective revisions that change the assigned token; and (3) Effective Revision Ratio := No. Eff. Revision/No. Total Revision.[2] The remaining ineffective revisions correspond to flip–flop events (Section 4).

**Baselines.** We evaluate our approach relative to standard greedy diffusion decoding and two training free revocable parallel diffusion decoding baselines, WINO [3] (Hong et al., 2025) and Saber [4] (Dong et al., 2025). For both baselines, we use the authors' original implementations and evaluate them under the same experimental settings as ours for a fair comparison.

### 6.2. Main Results

**Performance on Benchmarks.** Table 1 shows that COVER generally preserves or improves task performance across four benchmarks and multiple diffusion models while substantially reducing decoding steps. Across both code generation (HumanEval, MBPP) and math reasoning (GSM8K, MATH500), COVER achieves the strongest or near-strongest accuracy in each model and length setting, thereby avoiding the noticeable quality degradation

---

[2]For Saber and WINO, revisions correspond to REMASK operations. For COVER, revisions include both REMASK and REPLACE; REPLACE is counted as effective because it modifies the previously assigned token in place.

[3]WINO: https://github.com/Feng-Hong/WINO-DLLM

[4]Saber: https://github.com/zhaoyMa/Saber

*Table 1.* Main results across four benchmarks and multiple diffusion models. We report accuracy (%) and the average number of decoding steps (lower is better). Speed denotes relative runtime (baseline = 1.00×), where larger values are faster. Rows with a pink background indicate ours, and the best result within each block is **bolded**.

| Model | Len | Method | HumanEval | | | MBPP | | | GSM8K | | | MATH500 | | |
|---|---|---|---|---|---|---|---|---|---|---|---|---|---|---|
| | | | Acc.(%)↑ | Steps↓ | Speed↑ | Acc.(%)↑ | Steps↓ | Speed↑ | Acc.(%)↑ | Steps↓ | Speed↑ | Acc.(%)↑ | Steps↓ | Speed↑ |
| LLaDA-Base-8B | 256 | baseline | 33.54 | 256 | 1.00× | 40.80 | 256 | 1.00× | 70.58 | 256 | 1.00× | 30.60 | 256 | 1.00× |
| | | Saber | $32.93_{-0.61}$ | $93.75_{-162.25}$ | 2.67× | $41.20_{+0.40}$ | $107.11_{-148.89}$ | 2.34× | $69.62_{-0.96}$ | $135.36_{-120.64}$ | 1.72× | $31.80_{+1.20}$ | $123.91_{-132.09}$ | 2.02× |
| | | WINO | $33.54_{+0.00}$ | $71.43_{-184.57}$ | 2.19× | $41.80_{+1.00}$ | $82.08_{-173.92}$ | 1.55× | $70.43_{-0.15}$ | $106.76_{-149.24}$ | 1.09× | $30.80_{+0.20}$ | $102.30_{-153.70}$ | 1.16× |
| | | **COVER** | $\mathbf{34.76}_{+1.22}$ | $\mathbf{46.40}_{-209.60}$ | **2.98×** | $\mathbf{42.40}_{+1.60}$ | $75.37_{-180.63}$ | 2.42× | $\mathbf{70.96}_{-0.38}$ | $97.21_{-158.79}$ | 1.78× | $\mathbf{32.00}_{+1.40}$ | $\mathbf{87.66}_{-168.34}$ | **2.08×** |
| | 512 | baseline | 32.93 | 512 | 1.00× | 39.80 | 512 | 1.00× | 70.74 | 512 | 1.00× | 31.80 | 512 | 1.00× |
| | | Saber | $33.54_{+0.61}$ | $114.81_{-397.19}$ | 4.45× | $40.20_{+0.40}$ | $167.99_{-344.01}$ | 3.05× | $70.13_{-0.61}$ | $166.08_{-345.92}$ | 3.07× | $30.20_{-1.60}$ | $181.98_{-330.02}$ | 2.80× |
| | | WINO | $34.76_{+1.83}$ | $103.18_{-408.82}$ | 2.75× | $40.40_{+0.60}$ | $125.35_{-386.65}$ | 1.92× | $70.05_{-0.69}$ | $128.30_{-383.70}$ | 1.49× | $31.60_{-0.20}$ | $139.45_{-372.55}$ | 1.76× |
| | | **COVER** | $\mathbf{35.37}_{+2.44}$ | $\mathbf{70.90}_{-441.10}$ | **4.64×** | $\mathbf{41.00}_{+1.20}$ | $94.03_{-417.97}$ | 3.82× | $\mathbf{71.42}_{-0.68}$ | $105.15_{-406.85}$ | 3.17× | $\mathbf{32.40}_{+0.60}$ | $124.29_{-387.71}$ | 2.91× |
| LLaDA-Ins-8B | 256 | baseline | 39.63 | 256 | 1.00× | 37.20 | 256 | 1.00× | 74.91 | 256 | 1.00× | 30.60 | 256 | 1.00× |
| | | Saber | $40.24_{+0.61}$ | $122.55_{-133.45}$ | 2.09× | $37.60_{+0.40}$ | $114.86_{-141.14}$ | 2.25× | $75.66_{+0.75}$ | $120.48_{-135.52}$ | 2.47× | $31.80_{+1.20}$ | $130.66_{-125.34}$ | 2.25× |
| | | WINO | $39.63_{+0.00}$ | $88.64_{-167.36}$ | 2.89× | $36.20_{-1.00}$ | $87.00_{-169.00}$ | 1.64× | $\mathbf{77.33}_{+2.42}$ | $49.47_{-206.53}$ | 3.01× | $32.40_{+1.80}$ | $75.49_{-180.51}$ | 1.96× |
| | | **COVER** | $\mathbf{40.24}_{+0.61}$ | $96.16_{-159.84}$ | 2.66× | $\mathbf{39.00}_{+1.80}$ | $69.75_{-186.25}$ | 2.35× | $77.26_{+2.35}$ | $51.65_{-204.35}$ | **3.25×** | $\mathbf{32.80}_{+2.20}$ | $68.05_{-187.95}$ | **2.52×** |
| | 512 | baseline | 46.95 | 512 | 1.00× | 37.60 | 512 | 1.00× | 79.08 | 512 | 1.00× | 37.00 | 512 | 1.00× |
| | | Saber | $47.56_{+0.61}$ | $191.29_{-320.71}$ | 2.58× | $37.40_{-0.20}$ | $168.44_{-343.56}$ | 2.96× | $79.61_{+0.53}$ | $128.75_{-383.25}$ | 3.87× | $38.20_{+1.20}$ | $170.12_{-341.88}$ | 2.92× |
| | | WINO | $47.56_{+0.61}$ | $178.44_{-333.56}$ | 1.59× | $38.40_{+0.80}$ | $131.29_{-380.71}$ | 1.94× | $79.45_{+0.37}$ | $74.96_{-437.04}$ | 3.61× | $39.20_{+2.20}$ | $115.69_{-396.31}$ | 2.33× |
| | | **COVER** | $\mathbf{48.17}_{+1.22}$ | $132.63_{-379.37}$ | 2.72× | $\mathbf{38.80}_{+1.20}$ | $115.24_{-396.76}$ | 3.08× | $\mathbf{79.98}_{+0.90}$ | $65.47_{-446.53}$ | **5.52×** | $\mathbf{40.80}_{+3.80}$ | $99.97_{-412.03}$ | **3.62×** |
| LLaDA-1.5-8B | 256 | baseline | 40.24 | 256 | 1.00× | **38.60** | 256 | 1.00× | 82.11 | 256 | 1.00× | 35.20 | 256 | 1.00× |
| | | Saber | $40.24_{+0.00}$ | $125.23_{-130.77}$ | 2.04× | $37.60_{-1.00}$ | $124.42_{-131.58}$ | 2.38× | $81.43_{-0.68}$ | $107.32_{-148.68}$ | 2.33× | $35.80_{+0.60}$ | $132.51_{-123.49}$ | 2.34× |
| | | WINO | $40.24_{+0.00}$ | $92.08_{-163.92}$ | 2.78× | $37.80_{-0.80}$ | $89.99_{-166.01}$ | 1.65× | $81.12_{-0.99}$ | $85.03_{-170.97}$ | 2.44× | $34.60_{-0.60}$ | $96.73_{-159.27}$ | 2.05× |
| | | **COVER** | $\mathbf{42.07}_{+1.83}$ | $87.02_{-168.98}$ | **2.94×** | $37.80_{-0.80}$ | $70.52_{-185.48}$ | 2.45× | $81.73_{-0.38}$ | $63.24_{-192.76}$ | 2.65× | $\mathbf{36.00}_{+0.80}$ | $66.17_{-189.83}$ | **2.66×** |
| | 512 | baseline | 48.78 | 512 | 1.00× | 38.20 | 512 | 1.00× | 82.34 | 512 | 1.00× | 40.00 | 512 | 1.00× |
| | | Saber | $48.17_{-0.61}$ | $225.11_{-286.89}$ | 2.22× | $38.40_{+0.20}$ | $185.50_{-326.50}$ | 2.70× | $81.35_{-0.99}$ | $117.53_{-394.47}$ | 4.29× | $41.00_{+1.00}$ | $213.55_{-298.45}$ | 2.37× |
| | | WINO | $\mathbf{48.78}_{+0.00}$ | $205.16_{-306.84}$ | 1.39× | $38.00_{-0.20}$ | $139.51_{-372.49}$ | 1.84× | $82.03_{-0.31}$ | $74.73_{-437.27}$ | 3.72× | $41.40_{+1.40}$ | $122.32_{-389.68}$ | 2.59× |
| | | **COVER** | $\mathbf{48.78}_{+0.00}$ | $140.44_{-371.56}$ | 2.66× | $\mathbf{39.40}_{+1.20}$ | $123.48_{-388.52}$ | 2.83× | $\mathbf{82.56}_{+0.22}$ | $63.84_{-448.16}$ | **5.41×** | $\mathbf{42.60}_{+2.60}$ | $111.05_{-400.95}$ | 2.82× |
| Dream-Ins-7B | 256 | baseline | 54.88 | 256 | 1.00× | 56.80 | 256 | 1.00× | 75.82 | 256 | 1.00× | 40.00 | 256 | 1.00× |
| | | Saber | $54.88_{+0.00}$ | $186.16_{-69.84}$ | 1.45× | $57.00_{+0.20}$ | $207.17_{-48.83}$ | 1.31× | $71.04_{-4.78}$ | $203.55_{-52.45}$ | 1.34× | $42.00_{+2.00}$ | $172.76_{-83.24}$ | 1.56× |
| | | WINO | $54.88_{+0.00}$ | $83.76_{-172.24}$ | 2.70× | $57.00_{+0.20}$ | $57.08_{-198.92}$ | 4.23× | $74.00_{-1.82}$ | $83.00_{-173.00}$ | 3.01× | $42.40_{+2.40}$ | $139.64_{-116.36}$ | 1.77× |
| | | **COVER** | $\mathbf{55.49}_{+0.61}$ | $71.56_{-184.44}$ | 2.77× | $\mathbf{58.00}_{+1.20}$ | $42.35_{-213.65}$ | **4.36×** | $\mathbf{78.92}_{+3.10}$ | $52.11_{-203.89}$ | 3.51× | $\mathbf{43.60}_{+3.60}$ | $105.30_{-150.70}$ | **1.84×** |
| | 512 | baseline | 54.27 | 512 | 1.00× | 57.20 | 512 | 1.00× | 76.88 | 512 | 1.00× | 42.40 | 512 | 1.00× |
| | | Saber | $52.44_{-1.83}$ | $448.85_{-63.15}$ | 1.24× | $55.40_{-1.80}$ | $461.57_{-50.43}$ | 1.22× | $72.10_{-4.78}$ | $450.30_{-61.70}$ | 1.26× | $45.00_{+2.60}$ | $352.11_{-159.89}$ | 1.59× |
| | | WINO | $53.05_{-1.22}$ | $99.37_{-412.63}$ | 4.73× | $56.20_{-1.00}$ | $62.11_{-449.89}$ | 7.82× | $74.07_{-2.81}$ | $96.98_{-415.02}$ | 5.05× | $44.40_{-2.00}$ | $204.58_{-307.42}$ | 2.38× |
| | | **COVER** | $\mathbf{56.10}_{-1.83}$ | $69.72_{-442.28}$ | **5.91×** | $\mathbf{57.60}_{+0.40}$ | $35.84_{-476.16}$ | **11.64×** | $\mathbf{79.30}_{+2.42}$ | $51.48_{-460.52}$ | **8.48×** | $\mathbf{45.40}_{+3.00}$ | $143.91_{-368.09}$ | **2.95×** |

of naive multi-token unmasking. The gains are particularly clear on code benchmarks: for LLaDA-Base-8B, COVER improves HumanEval from 32.93% to 35.37% at length 512 and MBPP from 40.80% to 42.40% at length 256; for LLaDA-Ins-8B at length 256, it improves HumanEval from 39.63% to 40.24% and MBPP from 37.20% to 39.00%. We also observe improvements in several reasoning settings. For example, on Dream-Ins-7B at length 512, COVER reaches 79.30% on GSM8K and 45.40% on MATH500, exceeding the baseline and surpassing the other two revocable methods. Overall, these results indicate that COVER enables aggressive parallel drafting with faithful in place verification, generally preserving or improving quality rather than simply trading accuracy for speed.

**Efficiency and Decoding Speed.** Beyond accuracy, COVER substantially reduces the number of diffusion steps and delivers faster end-to-end decoding. Within each model block, the speedups are measured relative to the standard one token per step baseline (Speed = 1.00×), and COVER is consistently among the fastest methods while maintaining competitive or best accuracy. For LLaDA-Base-8B at length 256, COVER cuts HumanEval steps from 256 to 46.40 (2.98× speedup) and reduces MATH500 steps to

87.66 (2.08×). For LLaDA-Ins-8B at length 512, COVER achieves a 5.52× speedup on GSM8K with only 65.47 steps, while also improving accuracy. On Dream-Ins-7B, the acceleration is even more pronounced, reaching up to 11.64× speedup on MBPP at length 512. These efficiency gains support the central claim of COVER: by reusing cached representations to stabilise parallel drafting and verifying only a small set of high-risk positions, we improve net denoising progress per step and avoid spending steps on ineffective oscillations.

We provide additional robustness evaluations in the appendix. Appendix C and Table 4 study stochastic decoding with non-zero temperatures, while Appendix D and Table 5 evaluate COVER on the multimodal diffusion LM MMaDA with ScienceQA. Together, these results show that context-preserving verification remains effective beyond the main deterministic text-only setting.

### 6.3. Flip-Flop Oscillations: Empirical Analysis

Figure 3 summarizes flip-flop behaviour under deterministic and stochastic decoding. The detailed statistics are provided in Appendix Tables 6 and 7. For the ratio panel, we pool the effective and total revision counts over Hu-

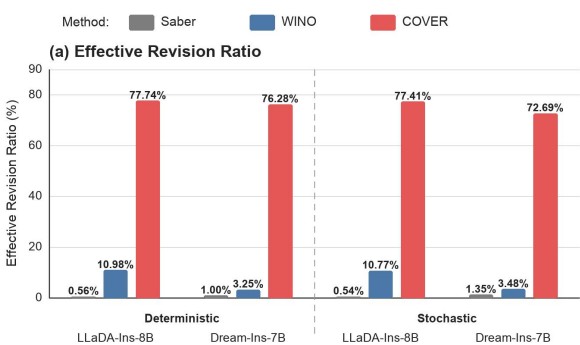

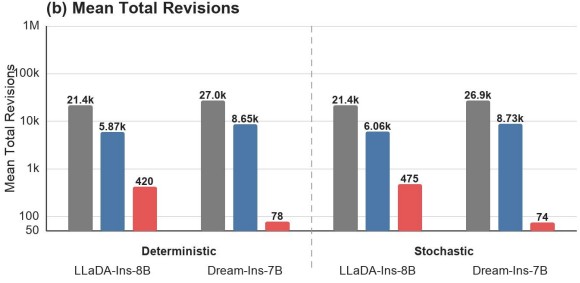

*Figure 3.* Flip-flop statistics under deterministic and stochastic decoding. The top panel reports the Effective Revision Ratio, and the bottom panel reports average dataset-level Total Revision counts over HumanEval and MATH500, with stochastic counts further averaged over temperatures. For Saber and WINO, revisions are REMASK operations; for COVER, revisions include both RE-MASK and REPLACE. The dashed line separates deterministic decoding from stochastic decoding with temperature sampling.

manEval and MATH500 before computing the ratio; for stochastic decoding, the counts are further pooled over temperatures $T \in \{0.1, 0.2, 0.3\}$. The count panel uses the same dataset-level averaging convention; we additionally report normalized per-sample counts below.

The results show that existing remask-based revocable decoders spend most of their revision budget on ineffective ReMask events. Under deterministic decoding, Saber obtains effective ratios of only $0.56\%$ on LLaDA-Ins-8B and $1.00\%$ on Dream-Ins-7B, indicating that nearly every Re-Mask operation is later undone by restoring the same token. WINO improves the ratio on LLaDA-Ins-8B to $10.98\%$, but remains similarly inefficient on Dream-Ins-7B with a ratio of $3.25\%$. In contrast, COVER achieves much higher effective ratios of $77.74\%$ and $76.28\%$ on the two models, respectively.

The same trend persists under stochastic decoding. COVER maintains effective ratios of $77.41\%$ on LLaDA-Ins-8B and $72.69\%$ on Dream-Ins-7B, whereas Saber remains near $1\%$ and WINO remains below $11\%$. The bottom panel further shows that this improvement does not come from performing more revisions: under deterministic decoding, COVER uses only $420/78$ average dataset-level revisions on LLaDA-

Ins-8B/Dream-Ins-7B, compared with $21.4k/27.0k$ for Saber and $5.87k/8.65k$ for WINO. Normalizing by sample count gives the same conclusion: deterministic COVER uses $1.27/0.23$ revisions per sample, compared with $64.46/81.27$ for Saber and $17.69/26.07$ for WINO. These results support our claim that context-preserving in-place verification avoids spending the unmasking budget on repeated flip-flop cycles, yielding higher revision effectiveness with substantially lower revision overhead.

### 6.4. Ablation Study

Table 2 ablates two core components of COVER on LLaDA-Ins-8B with generation length 256.

**Effect of KV cache override.** The variant **w/o kv** removes KV cache override and diagonal correction, and instead verifies by masking the seed positions in the input directly. This forces *all* queries to attend to a degraded context in which the recently drafted tokens are absent, so parallel drafting loses the conditioning signals it needs to remain stable. The consequence is immediate in both progress and runtime: the decoder revisits the same positions more often, spending iterations on low-value revoke and re-unmask cycles. Across all datasets, **w/o kv** substantially increases the step count and consistently slows inference. For example, on GSM8K, steps increase from 51.65 to 123.28, and speed drops to $0.63\times$; on HumanEval steps rise from 96.16 to 132.48 with a 0.61% accuracy drop. These results isolate KV cache override as the main mechanism that preserves a stable drafting context while still enabling faithful verification.

**Effect of stability aware seed selection.** The variant **w/o seed** keeps KV override verification but replaces stability aware seed selection with a naive confidence drop heuristic. While verification remains in place, the selected seeds are less compatible with cache reuse: the method more often verifies positions whose cached representations are likely to drift after the current draft, making the overridden memory less reliable as a conditioning context for other positions. Empirically, this primarily hurts efficiency rather than causing catastrophic failures. Across datasets, **w/o seed** increases the step count and reduces speed, for instance from 51.65 to 65.10 on GSM8K and from 96.16 to 102.14 on HumanEval, with smaller but consistent accuracy drops. This shows that seed selection is not merely a verification policy, but a prerequisite for making cache reuse robust under multi-token drafting.

### 6.5. Empirical validation of the drift proxy $d_{\text{out}}$

Our stability aware seed selection penalises candidates with large $d_{\text{out}}$, which serves as a proxy for how much their cached KV states may change after the current step update. To validate this proxy, we measure the true KV drift of each position as the average change in its key and value

*Table 2.* Ablation on LLaDA-Ins-8B with generation length 256. Speed is relative runtime (COVER = $1.00\times$).

| Dataset | Method | Acc.(%) ↑ | Steps ↓ | Speed ↑ |
|---|---|---|---|---|
| **HumanEval** | **COVER** | 40.24 | 96.16 | $1.00\times$ |
| | w/o kv | $39.63_{-0.61}$ | $132.48_{+36.32}$ | $0.73\times$ |
| | w/o seed | $39.02_{-1.22}$ | $102.14_{+5.98}$ | $0.79\times$ |
| **MBPP** | **COVER** | 39.00 | 69.75 | $1.00\times$ |
| | w/o kv | $38.60_{-0.40}$ | $147.85_{+78.10}$ | $0.68\times$ |
| | w/o seed | $38.00_{-1.00}$ | $82.78_{+13.03}$ | $0.92\times$ |
| **GSM8K** | **COVER** | 77.26 | 51.65 | $1.00\times$ |
| | w/o kv | $76.50_{-0.76}$ | $123.28_{+71.63}$ | $0.63\times$ |
| | w/o seed | $76.80_{-0.46}$ | $65.10_{+13.45}$ | $0.83\times$ |
| **MATH500** | **COVER** | 32.80 | 68.05 | $1.00\times$ |
| | w/o kv | $30.60_{-2.20}$ | $96.37_{+28.32}$ | $0.80\times$ |
| | w/o seed | $32.00_{-0.80}$ | $83.63_{+15.58}$ | $0.89\times$ |

*Table 3.* Compatibility of COVER with Fast-dLLM prefix cache. For COVER-only rows, Acc. and Steps match the corresponding length-256 COVER entries in Table 1, with Acc. reported as a fraction rather than a percentage. Speed is measured relative to COVER without prefix cache in each setting.

| Model | Dataset | Method | Acc. ↑ | Steps ↓ | Time (s) ↓ | Speed ↑ |
|---|---|---|---|---|---|---|
| Dream-Ins-7B | GSM8K | COVER | **0.7892** | 52.11 | 1.3521 | $1.00\times$ |
| | | +Prefix Cache | 0.7824 | **51.04** | **0.9345** | **$1.45\times$** |
| | MATH500 | COVER | **0.4360** | **105.30** | 2.8532 | $1.00\times$ |
| | | +Prefix Cache | 0.4060 | 107.04 | **2.0076** | **$1.42\times$** |
| LLaDA-Ins-8B | GSM8K | COVER | **0.7726** | **51.65** | 1.6521 | $1.00\times$ |
| | | +Prefix Cache | 0.7589 | 53.21 | **1.3469** | **$1.23\times$** |
| | MATH500 | COVER | **0.3280** | **68.05** | 2.2370 | $1.00\times$ |
| | | +Prefix Cache | 0.3200 | 77.24 | **1.9895** | **$1.12\times$** |

*Figure 4.* Spearman rank correlation between the proposed stability proxy $d_{\text{out}}$ and measured KV drift across diffusion models and tasks. Cell colour and the value indicate the correlation coefficient; values above $0.5$ suggest a strong monotonic relationship, supporting $d_{\text{out}}$ as a stability proxy.

states before versus after the update, averaged across layers and heads, and compute the Spearman rank correlation between $d_{\text{out}}$ and the measured drift (averaged over steps and examples). Figure 4 shows consistently positive correlations across all models and tasks (from $0.540$ to $0.716$, mean $= 0.637$); correlations above $0.5$ indicate a strong monotonic relationship, confirming that larger $d_{\text{out}}$ reliably corresponds to larger KV drift and supporting its use for avoiding unstable cache reuse.

### 6.6. Compatibility with Fast-dLLM

COVER reduces the number of decoding steps by avoiding ineffective revocation, while Fast-dLLM-style prefix caching primarily accelerates each decoding step by reusing prefix KV states. These two acceleration mechanisms target different parts of the generation cost. To verify their compatibility, we combine COVER with the prefix-cache acceleration proposed by Fast-dLLM (Wu et al., 2025).

As shown in Table 3, prefix caching can be added on top

of COVER to further improve wall-clock efficiency. It provides consistent speedups across Dream-Ins-7B and LLaDA-Ins-8B on GSM8K and MATH500, ranging from $1.12\times$ to $1.45\times$, while keeping accuracy within 3.0 percentage points of COVER-only decoding. These results suggest that COVER is complementary to cache-based dLLM acceleration: COVER improves net denoising progress by reducing flip-flop revisions, while prefix caching reduces the cost of each forward step.

## 7. Conclusion

DLLMs support parallel unmasking, but revocable decoding can waste computation through flip-flop oscillations that repeatedly remask tokens that would be restored unchanged. We introduce an in-place KV cache override verification mechanism with diagonal correction, enabling leave-one-out style checks while preserving a stable context for parallel drafting within a single forward pass. We also propose stability aware and adaptive seed selection that targets uncertain positions while avoiding unstable cache reuse, enabling efficient multi-token verification. Across benchmarks on different dLLMs, COVER generally preserves or improves accuracy while reducing decoding steps and inference time, delivering a better speed quality tradeoff than prior revocable methods.

## Impact Statement

COVER improves inference efficiency by reducing ineffective remasking in revocable diffusion decoding. Because it leaves model weights and capabilities unchanged, it introduces no new risks beyond those of the underlying dLLM.

## Acknowledgments

This work was supported in part by the UK Engineering and Physical Sciences Research Council through a Turing AI Fellowship (grant no. EP/V020579/1, EP/V020579/2) and the Prosperity Partnership scheme (grant no. UKRI566).

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

## A. Flip Flop Overhead and Step Lower Bound

Each flip flop at a position forces at least one additional unmask event beyond the first unmask of that position. Therefore, if $F$ is the total flip flop count, then the total number of unmask events is at least $L + F$. Since drafting selects at most $B$ positions per step, the total number of unmask events is at most $BT$, which yields Lemma A.1. This is a lower-bound statement: one additional flip flop need not change the ceiling immediately, but every $B$ additional flip flop events raise the lower bound by at least one step.

**Lemma A.1** (Unmask budget overhead from flip-flop). *Assume decoding terminates with no* [MASK] *tokens, and drafting unmasks at most $B$ positions per step, namely $|\mathcal{D}_t| \leq B$ for all $t$. Let $F$ be the total flip flop count defined above. Then the number of decoding steps satisfies*

$$T \geq \left\lceil \frac{L + F}{B} \right\rceil.$$

*Proof.* Let $n_i := |\mathcal{T}_i|$ be the number of times position $i$ is unmasked. Completion implies $n_i \geq 1$ for all $i$, hence $\sum_i n_i \geq L$. By construction, $F_i \leq n_i - 1$, so $n_i \geq 1 + F_i$ and thus $\sum_i n_i \geq L + \sum_i F_i = L + F$. Moreover, each unmask event corresponds to selecting one position into some $\mathcal{D}_t$, so $\sum_i n_i = \sum_{t=1}^{T} |\mathcal{D}_t| \leq BT$. Combining yields $BT \geq L + F$, giving the claim. $\qquad\square$

## B. Post-hoc diagonal correction for faithful verification

This appendix derives the closed form diagonal correction used in Sec. 5.1 to obtain faithful leave-one-out verification at seed queries without additional attention passes.

### B.1. Notation and objective

Consider a specific transformer layer $\ell$ and one attention head. Let $(Q_\ell, K_\ell, V_\ell)$ be computed from the verification input $\tilde{Y}^{(t-1)}$ where seed positions are masked. Let $(K'_\ell, V'_\ell)$ be the overridden memory formed by replacing the seed columns with their cached states from step $t - 1$. Define the overridden attention scores, weights, and outputs as

$$s_{i,j}^{\mathrm{ovr}} = \frac{q_i k'^{\top}_j}{\sqrt{d}},$$

$$w_{i,:}^{\mathrm{ovr}} = \mathrm{softmax}\left(s_{i,:}^{\mathrm{ovr}}\right),$$

$$o_i^{\mathrm{ovr}} = \sum_j w_{i,j}^{\mathrm{ovr}} v'_j,$$

where $q_i$ is the query at position $i$, and $(k'_j, v'_j)$ is the overridden key and value at position $j$.

For a seed query $i \in \mathcal{S}_{t-1}$, faithful verification requires restoring only the diagonal entry:

$$(k_j^{(i)}, v_j^{(i)}) = \begin{cases} (k_i, v_i), & j = i, \\ (k'_j, v'_j), & j \neq i, \end{cases}$$

where $(k_i, v_i)$ are the key and value from $(K_\ell, V_\ell)$ computed on the masked input. All off diagonal columns remain unchanged.

### B.2. Single score update lemma

**Lemma B.1** (Softmax under a single score change). *Let $w = \mathrm{softmax}(s) \in \mathbb{R}^L$. If we change only the $i$th score by $s_i \leftarrow s_i + \delta$, then the updated distribution $w'$ satisfies*

$$w'_j = \frac{w_j}{1 + w_i(\exp(\delta) - 1)} \quad \forall j \neq i,$$

$$w'_i = \frac{w_i \exp(\delta)}{1 + w_i(\exp(\delta) - 1)}.$$

*Proof.* Write $w_j = \exp(s_j)/Z$ where $Z = \sum_k \exp(s_k)$. After the change, $Z' = Z - \exp(s_i) + \exp(s_i + \delta) = Z\left(1 + w_i(\exp(\delta) - 1)\right)$. Substituting into $w'_j = \exp(s'_j)/Z'$ yields the claimed formulas. $\qquad\square$

### B.3. Closed form correction of attention weights

In our setting, restoring the diagonal key replaces only the diagonal score in row $i$:

$$\delta_i = \frac{q_i k_i^{\top}}{\sqrt{d}} - \frac{q_i k'^{\top}_i}{\sqrt{d}}.$$

Let $\alpha_i = w_{i,i}^{\mathrm{ovr}}$ be the overridden diagonal weight and define the scalar rescaling factor

$$r_i = 1 + \alpha_i\left(\exp(\delta_i) - 1\right).$$

Applying Lemma B.1 gives the corrected attention distribution for row $i$:

$$w_{i,j} = \frac{w_{i,j}^{\mathrm{ovr}}}{r_i} \quad \forall j \neq i, \qquad w_{i,i} = \frac{\alpha_i \exp(\delta_i)}{r_i}.$$

This shows explicitly that correcting the diagonal score changes the entire row through the shared normalizer.

### B.4. Closed form correction of attention outputs

The corrected output is

$$o_i = \sum_{j \neq i} w_{i,j} v'_j + w_{i,i} v_i.$$

Using the identities above and $o_i^{\mathrm{ovr}} = \sum_{j \neq i} w_{i,j}^{\mathrm{ovr}} v'_j + \alpha_i v'_i$, we obtain the single expression

$$o_i = \frac{o_i^{\mathrm{ovr}} - \alpha_i v'_i + \alpha_i \exp(\delta_i) v_i}{r_i}.$$

*Table 4.* Robustness of COVER under stochastic decoding. Baseline values are copied from the deterministic length-256 baseline rows in Table 1; stochastic COVER reports mean $\pm$ standard deviation over sampling temperatures $T \in \{0.1, 0.2, 0.3\}$.

| Model | Dataset | Method | Acc. (%) ↑ | Steps ↓ | Speed ↑ |
|---|---|---|---|---|---|
| Dream-Ins-7B | HumanEval | Baseline | 54.88 | 256.00 | 1.00× |
| | | COVER | 54.68 ± 0.35 | 74.74 ± 4.55 | (2.66 ± 0.17)× |
| | MATH500 | Baseline | 40.00 | 256.00 | 1.00× |
| | | COVER | 42.40 ± 0.87 | 108.54 ± 5.18 | (1.79 ± 0.09)× |
| LLaDA-Ins-8B | HumanEval | Baseline | 39.63 | 256.00 | 1.00× |
| | | COVER | 40.85 ± 2.20 | 96.30 ± 0.35 | (2.66 ± 0.01)× |
| | MATH500 | Baseline | 30.60 | 256.00 | 1.00× |
| | | COVER | 34.93 ± 0.99 | 67.30 ± 0.74 | (2.55 ± 0.03)× |

For non seed queries $i \notin \mathcal{S}_{t-1}$, no correction is applied and we keep $o_i = o_i^{\mathrm{ovr}}$.

## C. Robustness under Stochastic Decoding

The main experiments use deterministic decoding with temperature $T = 0$ to ensure reproducibility and controlled comparison. We further evaluate whether COVER remains effective under stochastic decoding, where token predictions are sampled from non-zero-temperature distributions. Specifically, we test $T \in \{0.1, 0.2, 0.3\}$ and report the mean and standard deviation across temperatures, together with the corresponding deterministic baseline from the main results.

As shown in Table 4, COVER remains stable under moderate stochasticity. The accuracy variation is modest overall, with three of four settings below one point and LLaDA-Ins-8B on HumanEval showing a larger but still bounded standard deviation. Compared with the main-table deterministic baseline, stochastic COVER reduces the average number of steps from 256 to 67.30–108.54 and achieves $1.79\times$–$2.66\times$ speedup, while maintaining comparable or better accuracy. These results suggest that the proposed seed selection and context-preserving verification remain useful when the decoding trajectory is stochastic rather than fully greedy.

## D. Results on Multimodal Diffusion LM

To examine whether COVER is limited to text-only diffusion language models, we further evaluate it on MMaDA (Yang et al., 2025), a multimodal diffusion language model, using ScienceQA (Lu et al., 2022), a multimodal science reasoning benchmark. Following the main evaluation protocol, we compare COVER with the original MMaDA decoder and WINO under the same decoding budget.

As shown in Table 5, COVER also improves the quality–efficiency trade-off on the multimodal diffusion model. Compared with the original MMaDA decoder, COVER improves accuracy from 47.99% to 51.46%, while reducing

*Table 5.* Results on MMaDA with ScienceQA. We report accuracy, average decoding steps, and relative speedup over the original decoder.

| Method | Acc. ↑ | Steps ↓ | Speed ↑ |
|---|---|---|---|
| Original | 47.99 | 256.0 | 1.00× |
| WINO | 50.12 | 41.1 | 5.59× |
| COVER | **51.46** | **29.4** | **8.83×** |

the average number of decoding steps from 256.0 to 29.4 and achieving an $8.83\times$ speedup. Compared with WINO, COVER further reduces the decoding steps from 41.1 to 29.4 and improves accuracy by 1.34 points. These results suggest that context-preserving verification is not restricted to the text-only setting, and can also benefit multimodal diffusion decoding.

## E. Detailed Flip-Flop Statistics

*Table 6.* Detailed flip-flop statistics under deterministic decoding. No. Eff. Revision counts effective revisions where the token changes, No. Total Revision counts all revision operations, and Ratio is computed as No. Eff. Revision / No. Total Revision. For Saber and WINO, revisions correspond to REMASK operations; for COVER, revisions include both REMASK and REPLACE.

| Model | Dataset | Method | No. Eff. Revision | No. Total Revision | Ratio ↑ |
|-------|---------|--------|-------------------|--------------------|---------|
| LLaDA-Ins-8B | HumanEval | Saber | 144 | 16081 | 0.90% |
| | | WINO | 246 | 3093 | 7.95% |
| | | COVER | 99 | 138 | 71.74% |
| | MATH500 | Saber | 95 | 26721 | 0.36% |
| | | WINO | 1044 | 8655 | 12.06% |
| | | COVER | 554 | 702 | 78.92% |
| Dream-Ins-7B | HumanEval | Saber | 459 | 30744 | 1.49% |
| | | WINO | 142 | 6660 | 2.13% |
| | | COVER | 24 | 34 | 70.59% |
| | MATH500 | Saber | 79 | 23221 | 0.34% |
| | | WINO | 420 | 10650 | 3.94% |
| | | COVER | 95 | 122 | 77.87% |

*Table 7.* Detailed flip-flop statistics under stochastic decoding with temperature sampling. We evaluate temperatures $T \in \{0.1, 0.2, 0.3\}$ on the available datasets. Columns use the same revision definition as Table 6.

| Model | Dataset | Temp. | Method | No. Eff. Revision | No. Total Revision | Ratio ↑ |
|---|---|---|---|---|---|---|
| LLaDA-Ins-8B | HumanEval | 0.1 | Saber | 137 | 16257 | 0.84% |
| | | | WINO | 252 | 3145 | 8.01% |
| | | | COVER | 108 | 153 | 70.59% |
| | | 0.2 | Saber | 143 | 15570 | 0.92% |
| | | | WINO | 259 | 3229 | 8.02% |
| | | | COVER | 107 | 149 | 71.81% |
| | | 0.3 | Saber | 141 | 16031 | 0.88% |
| | | | WINO | 253 | 3190 | 7.93% |
| | | | COVER | 118 | 165 | 71.52% |
| | MATH500 | 0.1 | Saber | 86 | 26733 | 0.32% |
| | | | WINO | 1087 | 8924 | 12.18% |
| | | | COVER | 540 | 714 | 75.63% |
| | | 0.2 | Saber | 87 | 26823 | 0.32% |
| | | | WINO | 1047 | 8973 | 11.67% |
| | | | COVER | 629 | 785 | 80.13% |
| | | 0.3 | Saber | 104 | 27051 | 0.38% |
| | | | WINO | 1018 | 8887 | 11.45% |
| | | | COVER | 705 | 885 | 79.66% |
| Dream-Ins-7B | HumanEval | 0.1 | Saber | 503 | 30718 | 1.64% |
| | | | WINO | 148 | 6961 | 2.13% |
| | | | COVER | 23 | 34 | 67.65% |
| | | 0.2 | Saber | 578 | 30625 | 1.89% |
| | | | WINO | 172 | 7257 | 2.37% |
| | | | COVER | 31 | 45 | 68.89% |
| | | 0.3 | Saber | 728 | 30774 | 2.37% |
| | | | WINO | 162 | 6845 | 2.37% |
| | | | COVER | 23 | 33 | 69.70% |
| | MATH500 | 0.1 | Saber | 94 | 23121 | 0.41% |
| | | | WINO | 415 | 10474 | 3.96% |
| | | | COVER | 98 | 127 | 77.17% |
| | | 0.2 | Saber | 118 | 23022 | 0.51% |
| | | | WINO | 440 | 10444 | 4.21% |
| | | | COVER | 77 | 106 | 72.64% |
| | | 0.3 | Saber | 154 | 22973 | 0.67% |
| | | | WINO | 488 | 10399 | 4.69% |
| | | | COVER | 70 | 98 | 71.43% |

