# OpenReview forum: "Stop the Flip-Flop: Context-Preserving Verification for Fast Revocable Diffusion Decoding"
_ICML.cc/2026/Conference — ICML 2026 regular_

### Official Review · Reviewer_rurh · 2026-03-07

**Soundness:** 3
**Presentation:** 3
**Significance:** 2
**Originality:** 2
**Overall Recommendation:** 4
**Confidence:** 2

**Summary:**

This paper studies a previously insufficiently emphasized problem in revocable parallel decoding of diffusion language models (dLLMs): flip-flop oscillations, where certain tokens are repeatedly remasked and then restored to the same original tokens, resulting in additional step overhead without substantial error correction benefits. To address this issue, the authors propose COVER, a single forward verification mechanism based on KV cache override: when verifying seed tokens, their input positions are set to \`[MASK]\`, while the cached KV states are retained for other queries, thereby achieving both leave-one-out style verification and stable drafting; at the same time, diagonal correction is used to avoid self-leakage, and stability-aware seed selection combined with adaptive revision rate is adopted to improve efficiency. Experiments on multiple dLLMs, mathematical reasoning, and code generation benchmarks show that COVER can reduce invalid remasks, decrease decoding steps, and bring higher inference speed while maintaining or even improving quality.

**Compliance With Llm Reviewing Policy:**

Affirmed.

**Ethics Expertise Needed:**

["Other Expertise"]

**Final Justification:**

Thank you for your response. Most of my issues have been resolved, and I will improve my score.

**Key Questions For Authors:**

1. Is flip-flop oscillation still equally significant in more dLLMs, different block sizes, different drafting thresholds, or non-greedy settings?
2. What is the individual contribution of diagonal correction to the final effect? Have ablations been done to remove correction and retain only KV override?
3. What are the additional computational and memory overheads of COVER? Reporting a more fine-grained wall-clock breakdown would help better understand the source of speedup.
4. The method is currently mainly verified on mathematical and code benchmarks; in more open-ended text generation tasks, is context-preserving verification still effective?

**Limitations:**

yes

**Strengths And Weaknesses:**

### Strengths

* The paper identifies a very specific and practically impactful problem in revocable diffusion decoding: flip-flop oscillations. This phenomenon is clearly defined, and Figure 1 on Page 1 and Table 2 on Pages 7–8 provide convincing empirical support.
* The method design of COVER is relatively complete. KV cache override keeps the context of non-seed queries stable, while diagonal correction ensures that seed verification is closer to faithful leave-one-out, which is an ingenious technical point. Figure 2 on Page 4 also provides an intuitive illustration of the process.
* Stability-aware seed scoring considers not only uncertainty but also downstream influence and cache drift proxy, with reasonable motivation. Moreover, Figure 3 on Page 8 provides verification of the positive correlation between \`d\_out\` and actual KV drift.
* The experiments cover multiple models, two length settings, and four benchmarks, with generally stable results. Table 1 shows that COVER often improves accuracy, reduces steps, and brings significant speedup simultaneously; for example, it reaches a maximum of 11.64× on Dream-Ins-7B.
* The ablation experiments are quite valuable. Table 3 on Page 8 indicates that KV override is the main source of contribution, and seed selection also stably improves efficiency.

### Weaknesses

* The method's contributions are mainly concentrated at the inference/decoding mechanism level, and the work is more inclined to system and algorithm engineering optimization rather than a new dLLM modeling framework, so the novelty is moderate.
* The paper regards flip-flop as the main bottleneck, but current evidence is mainly from a limited number of model types and tasks; it can be further verified whether this phenomenon is equally dominant in a wider range of dLLMs or under different decoding policies.
* Several components coexist in the method: KV override, diagonal correction, three-way revision, stability-aware seed selection, and adaptive revision rate. Although there are partial ablations, the decomposition of the interaction between each module is still not detailed enough.
* The speed indicator is relative runtime, normalized compared to the standard greedy baseline, which is reasonable, but it lacks more fine-grained wall-clock and memory overhead analysis, such as the additional cost of KV cache override and correction itself.
* The paper is currently mainly evaluated in the greedy temperature=0 scenario, and the performance of the method in higher sampling temperatures or more open-ended generation is still unclear.

---

> ### Author Rebuttal · Authors · 2026-03-31
>
> We appreciate the time and effort you've taken to review our work.
>
> > **W2 Q1:** Generalization of the flip-flop bottleneck across dLLMs and decoding policies.
>
> Table 1. Flip-flop statistics under deterministic decoding. Each cell reports total ReMask count, with effective ReMask ratio in parentheses, under deterministic decoding (T=0).
> | Model                | Dataset   | saber           | wino            | COVER      |
> |----------------------|-----------|----------------:|----------------:|----------------:|
> | LLaDA-Ins    | HumanEval | 16081 (0.90%)   | 3093 (7.95%)    | 138 (71.74%)    |
> |   | Math500   | 26721 (0.36%)   | 8655 (12.06%)   | 760 (77.63%)    |
> | Dream-Ins | GSM8K     | 96126 (1.86%)   | 12220 (7.81%)   | 696 (71.55%)    |
> | | MBPP      | 229203 (0.32%)  | 5981 (1.94%)    | 87 (68.97%)     |
>
>
> Table 2. Flip-flop statistics across temperatures and block sizes.
> | Factor      | Value | saber         | wino          | COVER        |
> |-------------|------:|--------------:|--------------:|-------------:|
> | Temperature |   0.1 | 16257 (0.84%) | 3145 (8.01%)  | 153 (70.59%) |
> |             |   0.3 | 16031 (0.88%) | 3190 (7.93%)  | 165 (71.52%) |
> | Block size  |    32 | 52771 (0.4%)  | 5078 (12.1%)  | 501 (79.8%)  |
> |             |   128 | 65925 (0.4%)  | 15299 (9.8%)  | 862 (77.0%)  |
>
>
> Table 3. Flip-flop statistics across revision thresholds. (saber omitted, as it has no threshold)
> | Threshold | wino         | COVER       |
> |----------:|-------------:|------------:|
> |       0.7 | 8655 (12.1%) | 714 (80.0%) |
> |       0.9 | 2608 (12.3%) | 389 (68.4%) |
>
> Tables 1-3 show a consistent flip-flop pattern across models and settings: remask-based baselines have low effective ReMask ratios, while COVER achieves much higher effectiveness with far fewer total ReMask events. The same qualitative pattern holds under stochastic decoding, different block sizes, and different drafting thresholds.
>
>
> > **W3 Q2:** More Fine-grained Ablation Study.
>
> Table 4. HumanEval ablation of COVER components.
> | Variant | Acc. | Steps | Speed |
> |---------|-----:|------:|------:|
> | Full | 41.46 | 53.68 | 1.00 |
> | w/o KV override + diagonal correction | 38.41 | 105.73 | 0.73 |
> | w/o three-way revision | 40.24 | 69.07 | 0.75 |
> | w/o scoring + adaptive revision rate | 39.63 | 70.49 | 0.78 |
>
> We group COVER into three coupled blocks: (i) KV override + diagonal correction, (ii) three-way revision, and (iii) scoring + adaptive revision rate. We do not ablate diagonal correction or adaptive revision rate alone, because diagonal correction is required for leave-one-out verification with KV override, and adaptive revision rate is defined on top of the scoring signal.
>
> On HumanEval, the full model performs best. Removing KV override + diagonal correction causes the largest drop, confirming that context-preserving verification is the core of COVER. Among the remaining components, three-way revision mainly improves efficiency, while scoring + adaptive revision rate contributes more to accuracy.
>
>
> > **W4 Q3:** Memory overhead and wall-clock analysis
>
> Table 5. Memory Consumption on GSM8K under Long-Context 20-Shot Prompting (50 samples)
> | Model              | Method   | Avg. Peak Memory (GB) |
> |-------------------|----------|-----------------------|
> | LLaDA-Instruct | ori      | 19.22             |
> |  | Cover | **20.56**               |
> | Dream-Instruct | ori      | 18.06            |
> |  | Cover | **19.69**               |
>
> On GSM8K with 20-shot 4K+ input length prompts, COVER adds only modest memory overhead (+7.0% on LLaDA-Instruct and +9.0% on Dream-Instruct). KV override + diagonal correction account for 6.2% of runtime, and seed selection adds 2.1%.
>
> > **W5:** Performance of stochastic decoding.
>
> Table 6. Sensitivity to Sampling Temperature
>
> | Model     | Dataset   | Acc.         | Steps         | Speed        |
> |-----------|-----------|--------------|---------------|--------------|
> | Dream-Ins | HumanEval | 54.68 ± 0.35 | 74.74 ± 4.55  | 2.66 ± 0.17  |
> | LLaDA-Ins |  MATH      | 34.93 ± 0.99 | 67.30 ± 0.74  | 2.55 ± 0.03  |
>
> We also evaluate stochastic decoding with temperature∈{0.1,0.2,0.3}. Within this range, COVER remains stable in both accuracy and efficiency across models and datasets.
>
>
> > **Q4:** Performance on open-ended text generation.
>
> **Response to Q5:**
>
> Table 7. Results on ScienceQA with MMaDA.
> | Approach | Accuracy (%) | Steps | Speedup |
> |----------|--------------|-------|---------|
> | Ori      | 47.99        | 256   | 1.00x   |
> | Wino     | 50.12        | 41.1  | 5.59x   |
> | COVER    | 51.46        | 29.4  | 8.83x   |
>
> We further evaluate COVER on MMaDA, a multimodal diffusion language model, using ScienceQA, a multimodal reasoning benchmark that requires the model to reason over the input context before generating the final answer. COVER outperforms both the original decoder and Wino in both accuracy and efficiency, reaching 51.46% accuracy with 29.4 steps and 8.83x speedup.

---

> > ### Author Rebuttal · Reviewer_rurh · 2026-04-03
> >
> > Thank you for your response. Most of my issues have been resolved, and I will improve my score.

---

> > > ### Author Response · Authors · 2026-04-03
> > >
> > > Thank you for raising the score and for your valuable suggestions. Your feedback has been instrumental in improving the quality of our submission.

---

### Official Review · Reviewer_2ejY · 2026-03-09

**Soundness:** 3
**Presentation:** 2
**Significance:** 2
**Originality:** 3
**Overall Recommendation:** 4
**Confidence:** 3

**Summary:**

This paper addresses the flip-flop oscillation problem in revocable parallel decoding for diffusion language models by proposing a context-preserving verification mechanism named COVER. Existing revocable decoding methods frequently reset tokens to masks and then restore them unchanged during verification, which not only wastes the computational budget but also degrades the context for parallel drafting. By combining key-value cache override and diagonal attention correction in a single forward pass, COVER enables dual-view computation, preserving a stable drafting context while achieving exact leave-one-out verification for seed queries. In addition, the paper proposes a stability-aware seed selection strategy that dynamically adjusts verification by considering uncertainty, downstream influence, and cache drift risk. Experiments show that COVER significantly reduces ineffective remasking operations across multiple benchmarks, achieving up to 11.64x speedup while maintaining generation quality.

**Compliance With Llm Reviewing Policy:**

Affirmed.

**Final Justification:**

The rebuttal has addressed most of my concerns with substantial additional experiments and analysis. I have raised my score accordingly.

**Key Questions For Authors:**

1. Specific data on memory consumption and system-level overhead: Can you provide a specific comparative analysis of peak memory usage between COVER and baseline methods like WINO? Especially in extremely long-context scenarios or large-batch inference, will the local key-value cache override mechanism trigger significant memory fragmentation or introduce additional storage overhead that hinders practical deployment?

2. Sensitivity analysis of drafting confidence threshold and high-temperature sampling: The experiment treats the drafting threshold as an adjustable parameter and is primarily tested under greedy decoding. How sensitive is the overall system speedup to this threshold and the sampling temperature? If the sampling temperature is significantly increased, is the stability-aware scoring mechanism's prediction of cache drift still accurate?

**Strengths And Weaknesses:**

**Strengths:**
1. Precise problem targeting with high engineering value: The paper astutely captures the widespread flip-flop oscillation phenomenon in existing revocable decoders. Through empirical analysis, it points out that the vast majority of remasking operations in baseline methods are ineffective. This finding directly hits the efficiency pain point of current parallel decoding.

2. Elegant reconstruction of the underlying attention mechanism: By utilizing key-value cache override combined with closed-form diagonal attention correction, COVER simultaneously accomplishes stable draft generation and faithful verification in a single forward pass, without introducing auxiliary shadow blocks or additional model calls. This local modification of the attention matrix is mathematically rigorous and highly innovative.

**Weaknesses:**
1. Lack of analysis on memory management and system-level overhead: Although COVER significantly reduces the number of model evaluations, the local key-value cache override mechanism inevitably requires low-level memory pointer management or additional tensor allocation. The paper does not provide a comparison of peak memory usage for COVER in actual deployment. Especially under long-context sequences, the overhead and memory fragmentation caused by such memory operations might offset some of the computational gains.

2. Insufficient discussion on sensitivity to drafting confidence thresholds and high-temperature sampling scenarios: Current experiments are mainly based on greedy decoding or zero-temperature sampling, and the drafting confidence threshold is a manually set hyperparameter. In high-temperature sampling scenarios involving more creative tasks, or when the threshold parameter is sub-optimally set, it lacks sufficient ablation evidence to prove whether the stability-aware scoring based on marginal probabilities can still reliably select verification seeds. It remains questionable whether the system's speedup ratio and flip-flop oscillation suppression rate will experience a catastrophic degradation.

---

> ### Author Rebuttal · Authors · 2026-03-31
>
> We appreciate the time and effort you've taken to review our work.
>
> > **W1 Q1:** Analysis of memory usage.
>
> Table 1: Memory Consumption on GSM8K under Long-Context 20-Shot Prompting
> | Model              | Method   | Avg. Peak Memory (GB) |
> |-------------------|----------|-----------------------|
> | LLaDA-Instruct | ori      | 19.22             |
> | | wino     | 29.94               |
> | | Cover | **20.56**              |
> | Dream-Instruct | ori      | 18.06            |
> | | wino     | 30.95               |
> | | Cover | **19.69**               |
>
> To address this concern, added a long-context memory study on GSM8K with 20-shot prompting, block size 64, and output length 256, where the prompt length exceeds 4K tokens. Table 1 reports the mean peak GPU memory over 50 examples.
>
> COVER remains close to the original decoder in peak memory, while staying substantially below WINO. In absolute terms, COVER adds only 1.34 GB and 1.63 GB over the original decoder on the two models, compared with 10.72 GB and 12.89 GB for WINO. The main reason is that COVER only needs to cache the KV states of seed positions, and the number of seeds is typically small, usually no more than 10 positions for each step. As a result, the additional memory overhead is limited and remains practically acceptable.
>
>
>
> > **W2 Q2:** Sensitivity to confidence thresholds and temperature sampling.
>
> Table 2: Sensitivity to the Drafting Threshold τ
> | Model     | Dataset   | Acc. | Steps | Speed |
> |-----------|-----------|------|-------|-------|
> | Dream-Ins | HumanEval | 54.47 ± 1.04 | 70.96 ± 2.65 | 2.80 ± 0.11 |
> | | MATH500   | 43.00 ± 0.99 | 107.56 ± 4.54 | 1.79 ± 0.08 |
> | LLaDA-Ins | HumanEval | 41.46 ± 0.00 | 72.22 ± 16.43 | 2.73 ± 0.62 |
> | | MATH500   | 32.47 ± 0.34 | 85.83 ± 15.02 | 2.03 ± 0.38 |
>
> Table 3: Sensitivity to Sampling Temperature
>
> | Model     | Dataset   | Acc.         | Steps         | Speed        |
> |-----------|-----------|--------------|---------------|--------------|
> | Dream-Ins | HumanEval | 54.68 ± 0.35 | 74.74 ± 4.55  | 2.66 ± 0.17  |
> | | MATH      | 42.40 ± 0.87 | 108.54 ± 5.18 | 1.79 ± 0.09  |
> | LLaDA-Ins | HumanEval | 41.66 ± 0.35 | 54.48 ± 0.79  | 3.46 ± 0.05  |
> | | MATH      | 34.93 ± 0.99 | 67.30 ± 0.74  | 2.55 ± 0.03  |
>
> We agree that the robustness of COVER to both the drafting confidence threshold and stochastic decoding settings should be evaluated explicitly. To address this concern, we add two sensitivity analyses.
>
> For Table 2, we evaluate the **drafting threshold τ** over {0.7,0.8,0.9}. The results show that COVER is not particularly sensitive to this hyperparameter: accuracy is consistently stable across all settings, with standard deviation no larger than 1.04 points, and it is even identical on LLaDA-Ins/HumanEval. The main effect of τ is on efficiency, as smaller τ leads to more aggressive drafting, but the overall conclusion remains unchanged for all tested values. This suggests that the marginal-probability-based stability score does not rely on delicate threshold tuning to identify effective verification seeds.
>
> For Table 3, we evaluate **stochastic decoding** with temperature \in {0.1,0.2,0.3}. We do not observe catastrophic degradation in either quality or efficiency within this range: accuracy standard deviation stays below 1.0 point across all settings, and speed standard deviation is at most 0.17. This indicates that COVER continues to provide consistent acceleration under non-zero-temperature sampling, and that the proposed seed-selection mechanism remains robust to moderate sampling noise.
>
> Overall, these results support the claim that COVER is reasonably robust to both threshold choice and stochastic decoding. We agree that evaluating more aggressive temperature settings would be valuable, and we will clarify in the revised paper that, while the current ablations already go beyond greedy/zero-temperature decoding and show stable behavior, a broader study of very high-temperature regimes is left to future work.

---

> > ### Author Rebuttal · Reviewer_2ejY · 2026-04-01
> >
> > Thanks for your detailed responses. My concerns are almost addressed and I would like to increase my score accordingly.

---

> > > ### Author Response · Authors · 2026-04-02
> > >
> > > Thank you for raising the score and for your valuable suggestions. Your feedback has been instrumental in improving the quality of our submission.

---

### Official Review · Reviewer_4ZBx · 2026-03-10

**Soundness:** 3
**Presentation:** 3
**Significance:** 3
**Originality:** 3
**Overall Recommendation:** 5
**Confidence:** 2

**Summary:**

This paper identifies flip-flop oscillations in revocable diffusion decoding, which wastes steps and weakens drafting context. The authors propose the COVER a;gorithm, a single-pass verification method using KV-cache override, diagonal correction, and stability-aware seed selection to preserve context while verifying tokens, and reports faster decoding with maintained or improved quality, with speedups up to 11x.

**Compliance With Llm Reviewing Policy:**

Affirmed.

**Final Justification:**

The rebuttal resolves my concerns, and I keep my evaluation as "accept".

**Key Questions For Authors:**

See the weaknesses part.

**Limitations:**

yes

**Strengths And Weaknesses:**

Strengths:

1. Soundness: The paper identifies a failure mode in revocable diffusion decoding, and the proposed method is tightly matched to that failure mode: context-preserving verification addresses the context loss from remasking, while KEEP/REPLACE directly targets ineffective remask cycles.

2. Presentation: The high-level story is clear, and the main message is easy to understand.

3. Significance: The paper is trying to solve a practically relevant problem. Dllm inference speed is a major bottleneck, and the paper targets inference-time acceleration without retraining.


Weaknesses:

1. Significance: The impact could be high within dllms. The practical importance depends on how broadly diffusion LLMs are adopted and whether similar cache smoothness assumptions hold in other diffusion-style decoders.

2. Soundness: Although the empirical evidence is good, the paper lacks the theoretical justification of the proposed method. The authors do not establish a strong end-to-end guarantee for when the single-pass override truly matches exact leave-one-out verification through the full transformer stack.

3. Practical relevance: The paper emphasizes speedups, but it would be more convincing with a fuller systems analysis of the throughput under batch serving. Whether the proposed algorithm improves the throughput is highly practically relevant in the real setting.

---

> ### Author Rebuttal · Authors · 2026-03-31
>
> We appreciate the time and effort you've taken to review our work.
>
> > **W1:** Significance and generalizability to other diffusion-style decoders.
>
> Table 1: Results on ScienceQA with MMaDA.
> | Approach | Accuracy (%) | Steps | Speedup |
> |----------|--------------|-------|---------|
> | Ori      | 47.99        | 256   | 1.00x   |
> | Wino     | 50.12        | 41.1  | 5.59x   |
> | COVER    | 51.46        | 29.4  | 8.83x   |
>
> We agree that our claim should be scoped carefully: our goal is not to claim universality across all diffusion-style decoders, but to show that the context-preserving verification mechanism is useful beyond the specific text-only setups in the main paper.
>
> To probe this, we additionally evaluate COVER on **MMaDA**, a **multimodal diffusion language model**, on **ScienceQA**. Using the same decoding setup as in the main paper, COVER achieves the best accuracy while requiring the fewest decoding steps:
>
> This additional result provides further evidence that the benefit of COVER is not limited to the original model/task combinations in the paper. We will revise the paper to make the scope of our claim more precise and avoid overclaiming generality. We will also connect this result more explicitly to the drift analysis in the main paper, where our stability proxy is consistently correlated with measured KV drift across model/task pairs.
>
> To further evaluate the practical significance of our method, we additionally conduct experiments on MMaDA, a multimodal large diffusion language model, over ScienceQA, a large-scale multimodal science question answering benchmark. Compared with the original decoding baseline (47.99% accuracy, 256 steps, 1.0x speed), Wino achieves 50.12% accuracy with 41.1 steps (5.59x speedup), while COVER further improves the result to 51.46% accuracy with only 29.4 steps (8.83x speedup). This evidence suggests that our method is not restricted to the original setup and can provide practical benefits on another diffusion-based model and benchmark.
>
> > **W2:** Single-pass override vs. exact leave-one-out verification
>
> Table 2: Agreement between single-pass leave-one-out verification and exact leave-one-out verification.
>
> | Metric | Value |
> |--------|-------|
> | Total seed verifications | 1,470 |
> | Verdict agreement | 99.18% |
> | Disagreement rate | 0.82% |
>
> We agree that the current Appendix B establishes a narrower exact result at the attention-row level than an unconditional full-stack equivalence theorem.
> we provide a direct empirical validation of whether our single-pass leave-one-out verification matches exact leave-one-out verification through the full transformer stack, the results are in Table 2.
>
> Specifically, we compare: (1) our single-pass verification, and (2) an exact leave-one-out baseline that performs an additional full forward pass with only the target seed masked. We then check whether the two methods produce the same verification decision for each seed.
>
> As shown in Table2, the two verifiers agree on **99.18%** of sampled seed decisions, indicating that the proposed single-pass verifier is a highly faithful approximation to exact leave-one-out verification in the regimes we tested. While this empirical result is not a replacement for an unconditional end-to-end theorem, it directly addresses the concern at the full-transformer level. In the revision, we will make this distinction explicit by separating: (i) the **formal exact attention-level result** in Appendix B, and (ii) the **empirical full-stack fidelity** shown above.
>
>
> > **W3:** Practical throughput under batch serving.
>
> **Response to W3:**
>
> Table 3. Batched decoding throughput (tokens/s) under different batch sizes.
> | Method     | bs=1  | bs=4  |
> |------------|------:|------:|
> | ori        | 43.99 | 56.00 |
> | saber      | 118.42 | 120.58 |
> | wino       | 139.16 | 141.54 |
> | COVER | 170.58 | 189.80 |
>
> We reports batched decoding throughput in tokens/s. As batch size increases from 1 to 4, throughput improves for the original decoder, while COVER remain substantially faster overall.

---

> > ### Author Rebuttal · Reviewer_4ZBx · 2026-04-01
> >
> > The rebuttal resolves my concerns, and I keep my evaluation as "accept".

---

> > > ### Author Response · Authors · 2026-04-02
> > >
> > > Thank you for your valuable suggestions. Your feedback has been instrumental in improving the quality of our submission.

---

### Official Review · Reviewer_HDA8 · 2026-03-13

**Soundness:** 3
**Presentation:** 3
**Significance:** 3
**Originality:** 3
**Overall Recommendation:** 5
**Confidence:** 3

**Summary:**

This paper addresses inefficiencies in revocable parallel decoding for diffusion large language models (dLLMs). The authors identify "flip-flop oscillations"—where tokens are remasked during verification only to be restored to the same token later—as a major source of wasted computation in existing methods like Saber and WINO. To address this, they propose COVER (Cache Override Verification for Efficient Revision), which enables in-place leave-one-out verification via KV cache override with a closed-form diagonal correction. This allows parallel drafting to proceed with stable context while verifying seed positions from previous steps. The method also incorporates stability-aware seed selection that balances uncertainty, downstream influence, and cache drift. Experiments on LLaDA and Dream models across code generation and mathematical reasoning benchmarks demonstrate that COVER reduces flip-flop events by orders of magnitude, achieving up to 11.64× speedup while maintaining or improving accuracy compared to greedy decoding.

**Compliance With Llm Reviewing Policy:**

Affirmed.

**Final Justification:**

**Final Justification**

This paper addresses the inefficiency of "flip-flop oscillations" in revocable parallel decoding for diffusion large language models (dLLMs). The authors propose COVER (Cache Override Verification for Efficient Revision), which employs a KV cache override mechanism with diagonal correction to enable context-preserving verification within a single forward pass, coupled with stability-aware seed selection to minimize wasteful remasking operations.

---

**Summary of Original Concerns**

In my initial review, I raised four primary concerns: (1) the memory overhead of maintaining cached KV states for long-context generation, (2) compatibility with existing KV cache compression techniques, (3) sensitivity to hyperparameters (drafting threshold and budget), and (4) the lack of discussion on applicability to continuous diffusion or flow-based models.

---

**Assessment of Rebuttal**

The authors have provided thorough, data-driven responses that fully address my technical concerns:

- **Memory Analysis (Table 1):** The rebuttal demonstrates that COVER incurs only modest overhead (+7-9% vs. baseline, -31-36% vs. WINO) even for 4K+ token contexts, as it only caches K/V for selected seeds (typically ≤10) rather than shadow sequences.

- **Compatibility (Table 3):** Empirical validation shows COVER integrates seamlessly with Fast-dLLM's prefix caching, achieving additional 1.45× speedup without accuracy degradation, confirming orthogonal compatibility with compression methods.

- **Hyperparameter Robustness (Table 4):** Sensitivity analysis across τ ∈ {0.7, 0.8, 0.9} and B ∈ {10, 15, 20} demonstrates stable accuracy with only moderate efficiency variations, alleviating deployment concerns.

- **Scope Clarification:** While acknowledging that extension to continuous diffusion is non-trivial, the authors appropriately position COVER within masked discrete diffusion and demonstrate broader applicability via MMaDA evaluation (Table 2).

---

**Dimensional Evaluation**

*Soundness (3 → 4):* The technical approach is rigorous and well-motivated. The KV cache override mechanism with closed-form diagonal correction (Lemma B.1) is elegant and theoretically sound. The rebuttal strengthens this by validating stability proxies (d_out) and demonstrating compatibility with existing acceleration techniques.

*Originality (3 → 4):* The identification of flip-flop oscillations as a dominant inefficiency (over 99% in Saber, ~90% in WINO) provides a novel diagnostic lens for revocable decoders. The in-place verification via cache override represents a distinct technical contribution compared to auxiliary block approaches (WINO) or confidence-drop heuristics (Saber).

*Significance (3 → 4):* The practical impact is substantial, with demonstrated speedups up to 11.64× on Dream-7B while maintaining or improving accuracy. The training-free nature ensures immediate applicability to existing dLLMs (LLaDA, Dream, MMaDA), making this a valuable building block for the community.

*Clarity (3):* The paper is well-structured with clear exposition of the problem, methodology, and empirical validation. The rebuttal further clarifies scope limitations and deployment considerations.

---

**Conclusion**

Overall,the authors outline a fundamental problem regarding flip-flop oscillations in revocable diffusion decoding, presenting COVER as an elegant solution that preserves context while enabling efficient verification. The rebuttal has successfully resolved all my technical concerns through rigorous empirical validation and theoretical clarification.

Overall,this article examines an importan oncept in parallel decoding for diffusion language models, demonstrating significant practical speedups while maintaining accuracy across multiple benchmarks and model architectures. The combination of technical innovation, thorough evaluation, and practical applicability warrants strong acceptance.

**Final Recommendation: Accept (5)**

**Key Questions For Authors:**

1. **Memory analysis:** What is the memory overhead of maintaining the KV cache override mechanism for verification? For long sequences (e.g., 4k+ tokens), does the memory cost become prohibitive compared to standard revocable decoding, and how might this be mitigated?

2. **Generalization:** The method assumes a masked discrete diffusion framework. Could the core ideas (cache override with diagonal correction) be adapted to continuous diffusion language models or flow-matching models that operate in continuous latent spaces rather than discrete token spaces?

3. **Interaction with KV cache compression:** Many recent dLLM acceleration methods focus on KV cache eviction or quantization (e.g., Sparse-dLLM, referenced in the paper). How does COVER interact with these compression techniques? Would aggressive KV cache eviction interfere with the stability of the verification process?

4. **Theoretical guarantees:** Lemma A.1 provides a lower bound on steps given flip-flops. Can you provide any theoretical characterization of when COVER guarantees fewer flip-flops than baseline methods, or is the improvement purely empirical?

**Limitations:**

While the authors have pointed out the training-free property of COVER, the paper would benefit from explicit discussion of:
1. the memory overhead of maintaining cached KV states for seed positions, particularly for long-context generation,
2. the computational cost of the diagonal correction step at scale.

**Strengths And Weaknesses:**

**Strengths:**
*   **Problem identification:** The identification of flip-flop oscillations as a dominant inefficiency (over 99% of remasking operations in Saber, ~90% in WINO) is well-motivated and provides a clear lens for evaluating revocable decoders beyond raw accuracy.
*   **Technical contribution:** The KV cache override mechanism with diagonal correction is elegant and technically sound, enabling faithful leave-one-out verification within a single forward pass without destabilizing parallel drafting. The closed-form correction (Lemma B.1) is a nice theoretical contribution.
*   **Empirical rigor:** The evaluation spans four models (LLaDA variants, Dream) and four benchmarks (HumanEval, MBPP, GSM8K, MATH500), with consistent improvements in both accuracy and speed. The ablation studies (Table 3) effectively isolate the contributions of KV override and stability-aware selection.
*   **Practical impact:** The training-free nature of COVER makes it immediately applicable to existing dLLMs, and the significant wall-clock speedups (up to 11.64×) demonstrate clear practical value.

**Weaknesses:**
*   **Scope limitations:** The method is specifically designed for masked discrete diffusion models. The applicability to continuous diffusion or flow-based language models is not discussed, which limits the generalizability of the approach.
*   **Memory overhead:** While the paper focuses on speed, the memory overhead of maintaining cached KV states for verification (particularly for long sequences) is not thoroughly analyzed. This could be a practical constraint for deployment.
*   **Comparison breadth:** The comparison is limited to other diffusion-specific methods (Saber, WINO). A discussion of how these approaches relate to speculative decoding or lookahead decoding in autoregressive models would help contextualize the results for the broader ML community.
*   **Hyperparameter sensitivity:** The method introduces several thresholds ($\tau_{\text{draft}}$, drafting budget $B$) and the adaptive revision rate mechanism. While the paper mentions tuning $\tau_{\text{draft}}$, a more systematic analysis of sensitivity to these hyperparameters would strengthen the work.

---

> ### Author Rebuttal · Authors · 2026-03-31
>
> We appreciate the time and effort you've taken to review our work.
>
> > **W2 Q1:** Memory analysis.
>
> **Table 1. Memory Consumption on GSM8K under Long-Context 20-Shot Prompting (50 samples)**
> | Model              | Method   | Avg. Peak Memory (GB) |
> |-------------------|----------|-----------------------|
> | LLaDA-Instruct | ori      | 19.22             |
> |  | wino     | 29.94               |
> |  | Cover | **20.56**              |
> | Dream-Instruct | ori      | 18.06            |
> |  | wino     | 30.95               |
> |  | Cover | **19.69**               |
>
> We conduct a long-context memory analysis on GSM8K using 20-shot prompting, block size 64, and output length 256, which yields input lengths exceeding 4K tokens.
>
> COVER adds only modest memory over the original decoder (+7.0% on LLaDA; +9.0% on Dream), while reducing peak memory by ~31%-36% vs. WINO. The reason is that COVER stores only the previous-step K/V of selected seed positions, instead of maintaining a shadow sequence or large auxiliary cache. In our settings the seed set is small (typically <=10), so the overhead remains manageable even for 4K+ contexts.
>
> > **W1 Q2: adaptation to continuous diffusion or flow-matching language models.**
>
> We agree this broader positioning is useful, and we will clarify it in the revision. COVER is designed for **masked discrete diffusion** models. Extending it to continuous diffusion / flow-matching LMs is non-trivial because there is no direct analogue of discrete remasking or token replacement; one would need to redefine both the verification target and the revision operation in latent space.
>
> Similarly, we view COVER as **complementary** to speculative/lookahead decoding rather than directly comparable. Those methods target **causal autoregressive** generation, where accepted prefix tokens are not later revoked. Our setting is **revocable parallel decoding** for diffusion LMs, where explicit remasking can remove useful bidirectional context and cause flip-flop oscillations.
>
> To demonstrate broader practical significance within the diffusion family, we additionally evaluate COVER on **MMaDA**, a multimodal large diffusion language model, on **ScienceQA**:
>
> **Table 2. Results on ScienceQA with MMaDA**
>
> | Approach | Accuracy (%) | Steps | Speedup |
> |---|---:|---:|---:|
> | Original | 47.99 | 256.0 | 1.00x |
> | WINO | 50.12 | 41.1 | 5.59x |
> | COVER | **51.46** | **29.4** | **8.83x** |
>
> While this is not evidence for continuous/flow-based models, it shows that COVER is not limited to the text-only settings in the main paper.
>
>
> > **Q3:** Interaction with KV cache based acceleration approach.
>
> **Table 3. Compatibility of COVER with Fast-dLLM Prefix Cache**
> | Model     | Dataset | Method         | Acc.  | Steps | Speed |
> |-----------|---------|----------------|-------|-------|-------|
> | Dream-Ins | GSM8K   | COVER          | 0.789 | 52.1  | 1.00  |
> |           |         | +Prefix Cache  | 0.782 | 51.0  | 1.45  |
> | LLaDA-Ins | MATH500 | COVER          | 0.328 | 68.1  | 1.00  |
> |           |         | +Prefix Cache  | 0.320 | 77.2  | 1.12  |
>
> We also combine COVER with the prefix-cache acceleration proposed in Fast-dLLM. The two methods are clearly compatible: adding prefix cache on top of COVER further improves speed from 1.00x to 1.45x on Dream-Ins/GSM8K and to 1.12x on LLaDA-Ins/MATH500, with broadly comparable accuracy.
>
> > **W4:** Hyperparameter Sensitivity (Drafting Threshold and Drafting Budget).
>
> **Table 4. Sensitivity to drafting threshold τ and max unmask budget B (mean ± std).**
> | Hyperparameter | Setting                  | Acc.           | Steps          | Speed         |
> |----------------|--------------------------|----------------|----------------|---------------|
> | `τ \in {0.7,0.8,0.9}` | LLaDA-Ins / HumanEval | 41.46 ± 0.00  | 72.22 ± 6.43  | 2.73 ± 0.62  |
> |                | Dream-Ins / MATH500      | 43.00 ± 0.99   | 107.56 ± 4.54  | 1.79 ± 0.08  |
> | `B \in {10,15,20}`       | LLaDA-Ins / HumanEval | 40.65 ± 0.93  | 55.43 ± 4.55  | 3.30 ± 0.26  |
> |                | Dream-Ins / MATH500      | 40.40 ± 0.20   | 91.63 ± 1.07   | 1.95 ± 0.05  |
>
> We further study the sensitivity to both the drafting threshold τ and the max unmask budget B. Overall, COVER is fairly robust to both hyperparameters: accuracy remains broadly stable, while efficiency varies only moderately within the tested ranges.
>
> > **Q4:** Theoretical guarantees.
>
> Lemma A.1 characterizes the overhead induced by flip-flops, rather than proving a universal dominance result. What we can characterize theoretically is the regime where COVER avoids them: under our revision rule, a seed is re-masked iff its leave-one-out confidence is below 𝜏_draft, so stable or confidently correctable seeds are updated in place (KEEP/REPLACE) rather than re-masked; stable seeds therefore cannot create a new flip-flop at that step. KV-cache override further preserves seed context for non-seed queries during verification.

---

> > ### Author Rebuttal · Reviewer_HDA8 · 2026-04-04
> >
> > **Acknowledgement:**
> > (a) Fully resolved - My concerns have been adequately addressed. If you select this option, please consider adjusting your score accordingly.
> >
> > ---
> >
> > **Reasons:**
> >
> > The authors have comprehensively addressed all my technical concerns through rigorous additional experiments and clarifications:
> >
> > 1. **Memory Overhead (W2 Q1):** Table 1 demonstrates that COVER adds only modest memory overhead (+7.0% on LLaDA, +9.0% on Dream) compared to the original decoder, while reducing peak memory by 31-36% versus WINO. This confirms feasibility for long-context scenarios (4K+ tokens), directly addressing my deployment concerns. The overhead remains manageable because COVER stores only previous-step K/V for selected seed positions (typically ≤10) rather than maintaining shadow sequences.
> >
> > 2. **Compatibility with KV Cache Compression (Q3):** Table 3 validates that COVER integrates seamlessly with Fast-dLLM's prefix caching, achieving additional speedups (up to 1.45×) without accuracy loss. This confirms the method's compatibility with existing acceleration techniques and alleviates concerns about interference with compression methods.
> >
> > 3. **Hyperparameter Sensitivity (W4):** The analysis in Table 4 shows COVER is robust to variations in drafting threshold τ ∈ {0.7, 0.8, 0.9} and budget B ∈ {10, 15, 20}, with stable accuracy and moderate efficiency variations. This mitigates concerns about practical usability and tuning difficulty.
> >
> > 4. **Scope and Generalization (W1 Q2):** While extending to continuous diffusion/flow-matching models remains non-trivial (as these lack discrete remasking analogs), the authors appropriately clarify this scope limitation. The additional evaluation on MMaDA (Table 2) demonstrates broader applicability within the diffusion family beyond text-only settings.
> >
> > 5. **Theoretical Characterization (Q4):** The explanation of how the KEEP/REPLACE/REMASK rule prevents flip-flops—remasking only when leave-one-out confidence is below τ_draft—provides clear theoretical intuition for why COVER avoids oscillatory behavior, complementing Lemma A.1's characterization of overhead.
> >
> > Overall,the authors outline a fundamental problem regarding flip-flop oscillations in revocable diffusion decoding, presenting COVER as an elegant solution that preserves context while enabling efficient verification.
> >
> > Overall,this article examines an importan oncept in parallel decoding for diffusion language models, demonstrating significant practical speedups (up to 11.64×) while maintaining or improving accuracy across multiple benchmarks and model architectures.
> >
> > Given these thorough responses and the strong empirical validation, I recommend upgrading my rating to **Accept (5)**.

---

> > > ### Author Response · Authors · 2026-04-04
> > >
> > > Thank you for raising the score and for your valuable suggestions. Your feedback has been instrumental in improving the quality of our submission.

---

### Decision · Program_Chairs · 2026-04-30

**Decision:**

Accept (regular)

**Comment:**

This paper defines and addresses the flip-flop oscillation problem in revocable parallel diffusion decoding by proposing COVER (Cache Override Verification for Efficient Revision). Flip-flop oscillation refers to an inefficient cycle where a token is remasked only to be restored to the same token; the authors observe that over 99% of existing methods (such as Saber and WINO) suffer from this issue. COVER combines KV cache override with closed-form diagonal correction to perform leave-one-out verification and stable drafting simultaneously within a single forward pass. By adding stability-aware seed selection—which integrates uncertainty, downstream influence, and cache drift—the framework achieves a speedup of up to 11.64×.

The paper identifies a practically significant problem in flip-flop oscillation and provides an elegant solution through the mathematically sophisticated mechanism of KV cache override and diagonal correction. The work is supported by substantial speedups of up to 11.64×, its training-free nature, and consistent performance across various models (LLaDA, Dream, MMaDA) and benchmarks. Following the rebuttal, all four reviewers had their concerns resolved and converged toward an Accept recommendation. Accordingly, the AC has decided to Accept the paper.